# Replisomes restrict SMC translocation in vivo

Qin Liao [1], Hugo B. Brandão [2], Zhongqing Ren[1,3] & Xindan Wang [1] ✉

Structural maintenance of chromosomes (SMC) complexes organize genomes by extruding DNA loops, while replisomes duplicate entire chromosomes. These essential molecular machines must collide frequently in every cell cycle, yet how such collisions are resolved in vivo remains poorly understood. Taking advantage of the ability to load SMC complexes at defined sites in the *Bacillus subtilis* genome, we engineered head-on and head-to-tail collisions between SMC complexes and the replisome. Replisome progression was monitored by genome-wide marker frequency analysis, and SMC translocation was monitored by time-resolved ChIP-seq and Hi-C. We found that SMC complexes do not impede replisome progression. By contrast, replisomes restrict SMC translocation regardless of collision orientations. Combining experimental data with simulations, we determined that SMC complexes are blocked by the replisome and then released from the chromosome. Occasionally, SMC complexes can bypass the replisome and continue translocating. Our findings establish that the replisome is a barrier to SMC-mediated DNA-loop extrusion in vivo, with implications for processes such as chromosome segregation, DNA repair, and gene regulation that require dynamic chromosome organization in all organisms.

Structural Maintenance of Chromosomes (SMC) complexes play a key role in shaping the three-dimensional organization of genomes from bacteria to humans[1–3]. The SMC complex binds to DNA, captures a small DNA loop, and enlarges the loop progressively by reeling the flanking DNA regions into the loop[4–7]. This process, known as DNA-loop extrusion, is a conserved mechanism for SMC actions. Through loop extrusion, SMC complexes generate interactions of DNA segments, which have been shown to contribute to many processes such as chromosome organization and segregation, gene expression, DNA replication, DNA recombination, and repair[8,9]. In cells, SMC complexes are capable of traversing thousands to millions of DNA base pairs during loop extrusion and will inevitably encounter other DNA-bound molecules on the crowded chromatin fiber.

One crucial machinery that the SMC complex encounters is the replisome, which duplicates entire chromosomes one nucleotide at a time. In eukaryotes, the SMC cohesin complexes load on the chromosomes during interphase and compact the DNA into topologically associating domains[10–12]. During DNA replication, cohesin complexes establish sister chromatid cohesion, which is important for proper chromosome segregation[13–15]. Thus, cohesins and replisomes collide frequently during replication[16–19]. However, the in vivo consequence of cohesin-replisome collisions and the effect of these collisions on DNA replication and chromosome folding are unknown. One major challenge to investigate these problems in eukaryotes in vivo is the lack of defined sites for replisome-cohesin collisions. This complexity motivated us to use a bacterial model, which has defined replication origin, defined SMC loading sites, and definable replisome-SMC collision sites, to dissect the fundamental principles governing the rules of engagement[20] between SMC and the replisome in cells.

Across the bacterial domain of life, there are four classes of SMC complexes: the SMC-ScpAB complex, MukBEF complex, MksBEF complex, and WadjetABCD (MksBEFG) complex[21–25]. The SMC-ScpAB complex is widely distributed in bacteria and has the strongest homology to eukaryotic SMC complexes among the four classes of

[1]Department of Biology, Indiana University, Bloomington, IN, USA. [2]Illumina Inc., San Diego, CA, USA. [3]Present address: Molecular Biology Program, Memorial Sloan Kettering Cancer Center, New York, NY, USA. ✉e-mail: xindan@iu.edu

SMCs[1]. Our experimental model bacterium, *Bacillus subtilis*, only has the SMC-ScpAB complex. Therefore, in this study, we refer SMC-ScpAB complex as the SMC complex for simplicity.

In bacteria, DNA replication starts at a single origin and proceeds bidirectionally to the terminus region. Throughout the replication cycle, the SMC complex preferentially loads onto the *parS* sites by the partitioning protein ParB[21,26–28]. Once loaded, SMC travels processively from *parS* to the terminus and "zips" the left and right chromosomal arms[29]. This process connects DNA *in cis* and helps separate the newly replicated sister chromosomes[30,31] (Fig. 1a). Most bacterial genomes possess multiple *parS* sites, and they all have varying distances from the replication origin[32]. Because SMC and the replisome start their movement at different sites, collisions will occur frequently: SMC and the replisome traveling toward each other will generate a head-on

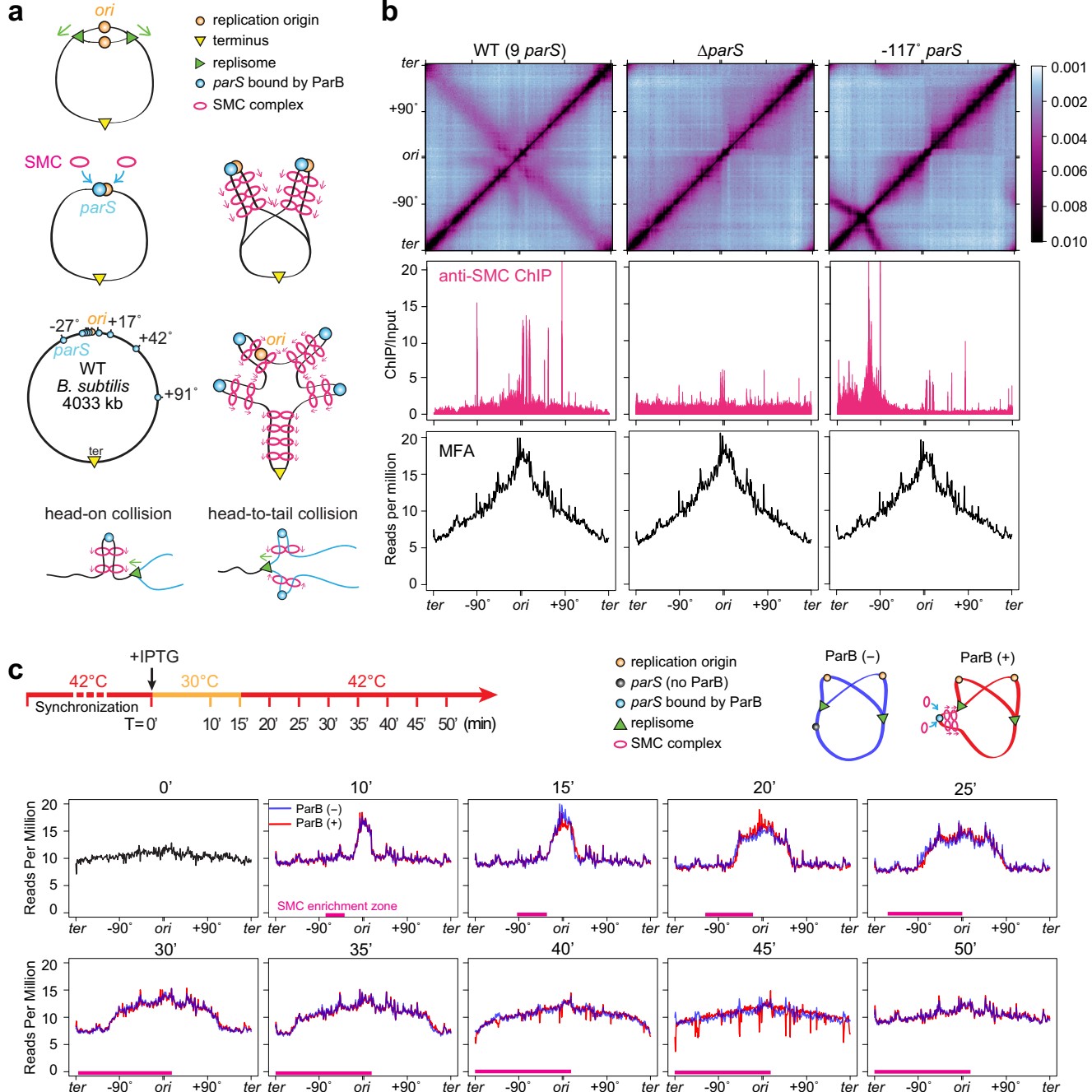

**Fig. 1 | SMC translocation does not affect replication progression.** See also Supplementary Figs. 1 and 2. **a** Schematics depicting replisome progression (top row), SMC loading and translocation helping origin segregation (2nd row), location of *parS* sites in WT *B. subtilis* and their effect on SMC loading (3rd row), and head-on and head-to-tail collisions between SMC and the replisome (bottom row). **b** Hi-C maps (top row), ChIP enrichments of SMC (middle row) and marker frequency analysis (MFA) plots (bottom row) of exponentially growing *B. subtilis* strains containing wild-type *parS* sites (PY79, left)[35], no *parS* site (BWX3212, middle)[29], or a single *parS* site at −117° (BWX3381, right)[29]. **c** Time-course MFA plots of a *dnaB*(ts) strain containing a single *parS* site at −59° and an IPTG-inducible copy of *parB* as the sole source of SMC loader protein (BWX5297). As indicated in the experimental timeline (top row), cells were arrested in G1 by growing at a non-permissive temperature (42 °C) for 45 min, then shifted to a permissive temperature (30 °C) for 15 min, and finally shifted back to 42 °C for the rest of the time to allow only one round of synchronized replication. See more analyses in Supplementary Fig. 1. Cells were grown with (red) or without (blue) IPTG at the onset of replication (T = 0 min). SMC enrichment zones determined by ChIP-seq plots (Supplementary Fig. 2b) are indicated by magenta bars.

collision; SMC and the replisome traveling in the same direction may generate a head-to-tail collision. Failures to resolve these collisions may have severe consequences on DNA replication and segregation. For instance, SMC could stall or collapse replication forks and perturb DNA replication; premature unloading of SMC by the replisome could impair chromosome segregation. Therefore, SMC-replisome collisions may have strong consequences for genome integrity, but the rules of engagement between SMC and the replisome in bacterial cells remain unclear.

Here, we investigate the collisions between the SMC complex and the replisome in vivo. We have taken advantage of the *B. subtilis* system, which much better tolerates the manipulations of SMC loading than other bacterial species[29,33,34]. We have controlled SMC loading spatially by changing the location of SMC loading sites (*parS*), and temporally by inducing the expression of the SMC loader (ParB) at desired times. Combined with our ability to control the initiation and progression of DNA replication using mutants and drugs, we have assessed the fate of SMC complexes and the replisome when they collide in a head-on or a head-to-tail orientation. Experimentally, we have tracked the changes in chromosome organization, SMC distribution, and DNA replication over time using chromosome conformation capture (Hi-C) assays, chromatin immunoprecipitation with deep sequencing (ChIP-seq) assays, and genome-wide replication profiling. Then, we combine experimental data with simulations of SMC translocation and replisome movement to determine whether SMCs disassociate, stall, continue translocating, or exhibit a combination of these activities when encountering the replisome.

## Results

### SMC distribution does not affect the replication profile

We first investigated the effect of SMC occupancy on replisome progression using experiments. Since SMC is loaded by ParB binding to *parS*, strains containing different *parS* sites have varied SMC occupancy and chromosome folding patterns. We chose three strains: (1) the wild-type *B. subtilis* strain containing 9 *parS* sites, which had a gradient of SMC localization from the origin to the terminus by ChIP-seq, and had a secondary diagonal and a star-shaped interaction pattern near the origin by Hi-C[35]; (2) a strain lacking all 9 *parS* sites, which had uniform SMC distribution along the genome, and lacked a secondary diagonal on the Hi-C map[29]; (3) a strain harboring a single *parS* site at −117°, which had SMC concentrated within one chromosome half with less than half of the chromosomal DNA zipped as observed by Hi-C[29] (Fig. 1b). To understand whether the different SMC occupancy in these three strains affects replication progression in exponentially growing cells, we performed whole-genome sequencing (WGS) coupled with marker frequency analysis (MFA) to examine the genome-wide DNA replication profiles. Our experimental results showed that these three strains had nearly identical replication profiles, indicating that the distribution and movement of SMC have no effect on replication progression (Fig. 1b).

### Synchronizing replisome progression

To directly test the effect of SMC translocation on the progression of the replication fork, we synchronized DNA replication using a *dnaB*(ts) allele in our experiments[35,36]. Cells were first grown exponentially at a permissive temperature (30 °C), then shifted to a restrictive temperature (42 °C) for 45 min. This step allowed DNA replication to finish but inhibited new rounds of replication, arresting cells in G1 phase. Next, cells were shifted down to 30 °C to initiate replication synchronously. We incubated the cells at 30 °C for only 15 min, then shifted them back to 42 °C. This step allowed the current round of replication to progress, but prevented further initiation events, resulting in a single round of synchronized replication. We took samples at the indicated time points and used MFA to analyze replication progression

(Fig. 1c). In all the experiments mentioned in this study, we used the same procedure for the replication synchronization.

The MFA plots obtained from experiments showed that cells achieved reasonable synchrony for replication initiation, progression, and termination (Fig. 1c, blue lines). However, there were features indicating that the synchronization was not perfect (Supplementary Fig. 1a). First, at T = 0 before initiation, instead of a flat line, there was a shallow slope from the origin to the terminus (Fig. 1c at 0 min before replication initiation and Supplementary Fig. 1a, slope 1), indicating that a fraction of cells was not arrested in G1 but had pre-existing replisomes on the chromosome. Secondly, during replication, the peak height was lower than twofold of the baseline (Supplementary Fig. 1a, middle panel), indicating that not all the cells in the population had initiated replication, resulting in a small fraction of cells without replisomes. Thirdly, instead of a vertical line connecting the replicated region to the unreplicated region, there was a slope in the experimental profile (Supplementary Fig. 1a, slope 2), indicating that cells started replication within a narrow time window rather than at a precise time. Finally, after the synchronized round of replication was completed, the replication profile had a slope (Supplementary Fig. 1a, slope 3), indicating that replisomes were still present in a small fraction of cells.

To understand the replisome progression in our experiment, we performed simulations of replisome dynamics to reproduce the distribution of replisomes according to the MFA plots (see details in Supplementary Methods). We varied five parameters of replisome behavior, including the fraction of cells containing replisomes before initiation (e.g., pre-existing replisomes), the average time for a replisome to load to the origin, the speed of replisome movement at 30 °C and 42 °C, and the rate of spontaneous replisome stalling after replication. We identified one set of numbers that reproduced the MFA profiles for all the time points in this experiment (Supplementary Fig. 1b, c), as well as all the MFA plots in later figures (Supplementary Fig. 1d–f). Our simulations estimated that the fraction of pre-existing replisomes was ~18%, the speed of DNA replication at 42 °C was ~66 kb/min, and the calculated fraction of cells that initiate replication was ~95% (Supplementary Fig. 1b, also see details in Supplementary Methods).

### SMC translocation does not affect replisome progression

To directly test whether SMC translocation affects replisome progression using experiments, we induced SMC loading and translocation in replication-synchronized cells and monitored the difference in replication fork progression with and without translocating SMC (Fig. 1c, red and blue curves, respectively). The location of SMC loading was specified by a single *parS* site at −59°, and the timing of SMC loading was controlled by IPTG-inducible *parB*. We added IPTG at the onset of replication initiation. ChIP-seq results showed that SMC was loaded and enriched progressively as expected (Supplementary Fig. 2b). Judged by the SMC enrichment zone (Fig. 1c, magenta bars determined by Supplementary Fig. 2b) and replisome progression (Fig. 1c, red and blue curves), SMC-replisome collisions occurred from 15 min to 50 min. Nonetheless, in the presence of SMC, the replication progression profiles overlapped with ones in the absence of SMC (Fig. 1c, red and blue curves). These experimental results indicate that SMC translocation does not affect replication progression.

### SMC translocation is affected by replisomes

Conversely, we investigated the impacts of replisome movement on SMC translocation using experiments (Fig. 2 and Supplementary Movie 1). We used a strain that retained the endogenous *parS* at −27° but lacked the other eight *parS* sites. This strain also contained a *dnaB*(ts) allele to allow for synchronization of DNA replication using the procedure described above (Fig. 2a, MFA plots) but had ParB and the SMC complex expressed from their endogenous promoters. At

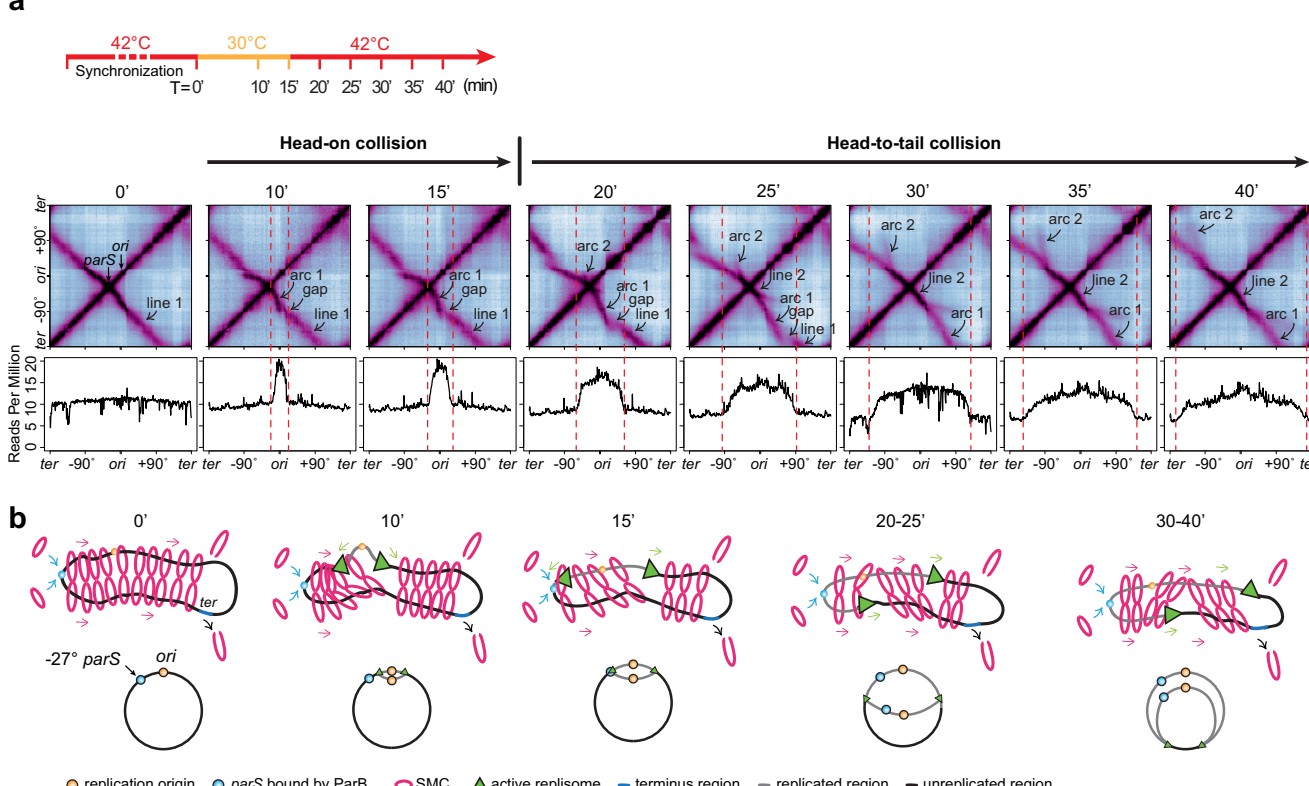

**Fig. 2 | SMC translocation is affected by replisomes.** See also Supplementary Fig. 3 and Supplementary Movie 1. **a** Time-course Hi-C maps and MFA plots of a strain containing a single *parS* site at −27° (BWX5230). G1-arrested cells were grown as indicated in the timeline (top row). T = 0′ indicates the onset of replication initiation. Samples were collected at the indicated time points. **b** Schematics depicting major SMC-replisome engagements at the indicated time points (top row). For simplicity, only one copy of the replicated region is shown (gray). Replisome progression is indicated on the bottom row.

T = 0, cells were arrested in G1 as indicated by the MFA plot. The Hi-C map had a secondary diagonal emanating from −27°, showing that the DNA regions flanking the *parS* were zipped (Fig. 2a, b, T = 0, line 1). From 0 min to 15 min, replication initiated and progressed toward the *parS* site. The replisome collided head-on with SMCs. We observed three new features on the Hi-C maps. First, starting at −27°, the secondary diagonal curved downward (Fig. 2a, arc 1), indicating that SMCs colliding with the replisome moved slower than the ones translocating toward the terminus. Secondly, there was a gap on the secondary diagonal (Fig. 2a, gap), which could be explained by SMC complexes being delayed at the replisome relative to those complexes that were already in motion ahead of the replisome. The delay could be caused by SMC being blocked or dissociated when colliding with the replisome (Fig. 2b, T = 10′−15′). Thirdly, the remaining portion of the secondary diagonal toward the terminus region (Fig. 2a, T = 10′−15′, line 1) maintained its shape/curvature seen at T = 0, indicating that SMCs running ahead of replication forks were unaffected by the replication. It is important to note that arc 1 extended beyond the leftward replisome, which became more evident at later time points. This observation indicates that some SMCs bypassed the replisome and continued translocation.

At 20−35 min, the replication fork moved beyond the *parS* site as indicated by MFA (Fig. 2a, T = 20′−25′). Thus, the SMCs loaded at the *parS* on the newly replicated DNA would trail behind the replisome and might collide with the leftward replisome in a head-to-tail manner (Fig. 2b, T = 20′−25′). The three features discussed above remained despite some minor changes (see schematics in Supplementary Fig. 3): (1) arc 1 continued to grow, and the growing tip of arc 1 still curved down near the replication fork; (2) the gap moved; (3) line 1 retreated toward the terminus. Additionally, we found that the central portion of

arc 1 (illustrated as line 2 in Supplementary Fig. 3), which was close to the *parS*, flattened out to a similar curvature as seen at T = 0, indicating that newly loaded SMCs were not affected by the replisome far ahead. In addition to these three features, we found a new, smaller arc (arc 2) emanating from the *parS* (−27°). Arc 2 curved up and tracked behind the replication fork. Our previous work indicates that each SMC complex (loop-extruding unit) contains two independent motors translocating away from the loading site[29,37–39]. The shape of arc 2 indicates that the SMC motor heading toward the terminus region traveled more slowly than the motor heading toward the replication origin. This is consistent with the idea that SMCs loaded behind the replisome are slowed down by the replisome, whereas the other SMCs translocating toward the origin region (i.e., away from the replisome) are less affected.

These experimental results demonstrate that when encountering the replisome, SMC movement was altered. The Hi-C maps show hints of SMC paused by the replisome (Fig. 2a, arc 1 and gap), and SMC bypassing the replisome (Fig. 2a, 10−30 min after replication initiation, arc 1 extending beyond the leftward replisome). It is also possible that some SMCs dissociated after the collisions. Given the continuous loading of SMCs, continuous movement of replisomes, and concurrent head-on and head-to-tail collisions between SMCs and replisomes, it is difficult to quantify the consequences of SMC-replisome collisions in this experimental setup. In our next sections, we set up simpler scenarios of SMC-replisome collisions to dissect the rules of engagement between SMC and the replisome.

### Engineering an in vivo system for SMC-replisome collisions
To better resolve the effect of the replisome on SMC translocation, we designed an experimental system to exert both temporal and spatial

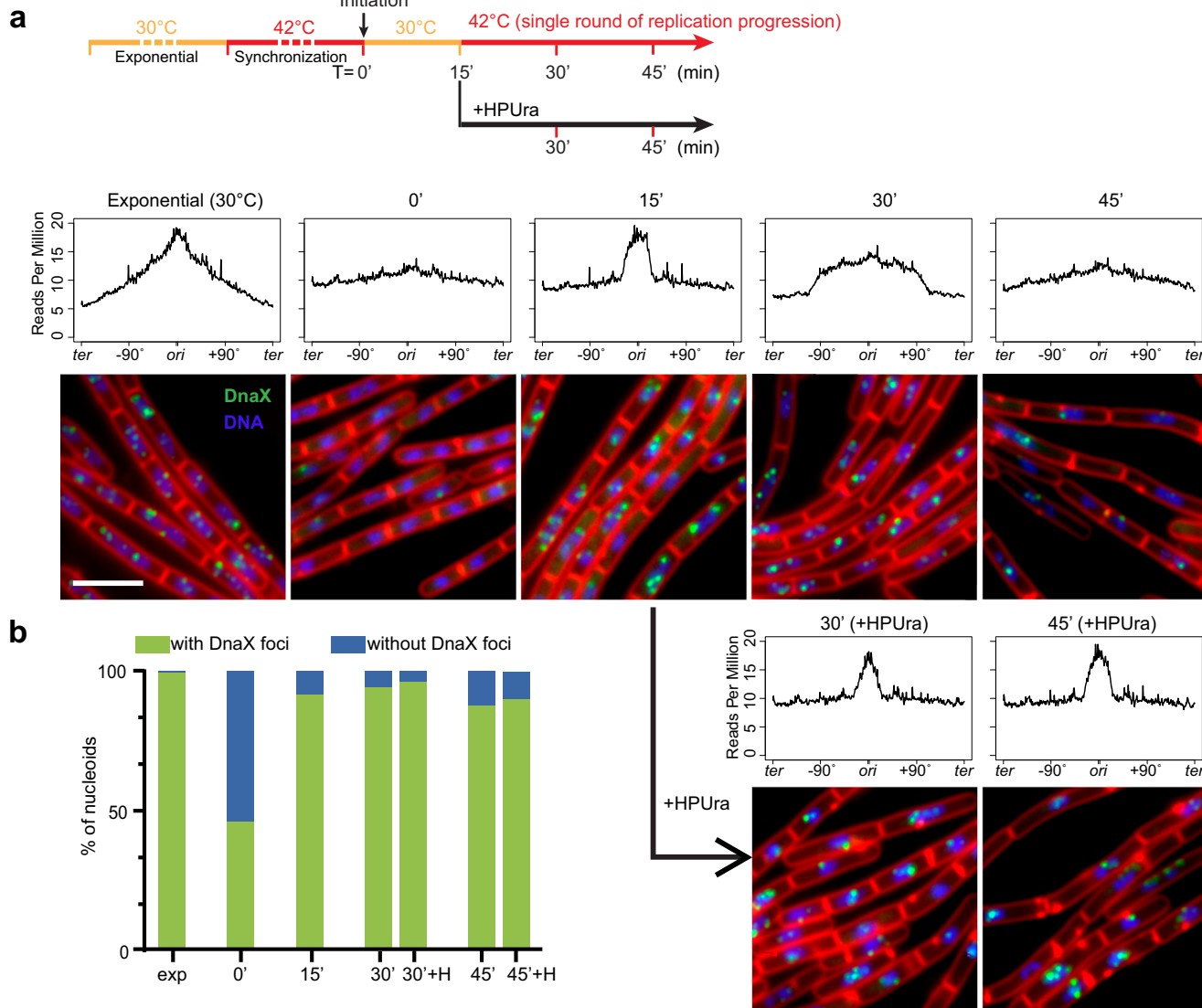

**Fig. 3 | HPUra stalls replication but does not affect replisome assembly.** See also Supplementary Fig. 4. **a** As shown in the experiment timeline (top row), a *dnaB*(ts) strain (BWX4310) was grown exponentially at 30 °C for 2 h, shifted to 42 °C for 45 min to synchronize DNA replication. At T = 0', cells were shifted to 30 °C for 15 min to initiate replication, then back to 42 °C to allow a single round of replication. HPUra was added at 15 min after replication initiation (T = 15'). Middle panel:

MFA plots are shown for the indicated time points. Bottom panel: representative images of a *dnaB*(ts) strain (BWX2533) grown as described above. The images contain DAPI-stained nucleoid (blue), FM-4-64-stained membrane (red), and YFP-tagged DnaX (green). Scale bar represents 4 μm. **b** Analysis of imaging results in (**a**). Nucleoids with or without DnaX foci were quantified at the indicated time points, and the percentages were plotted. Source data are provided as a Source Data file.

controls over the collisions between SMC and the replisome on the chromosome. For SMC, we inserted a single *parS* site at different genome positions to control SMC loading spatially and an IPTG-inducible *parB* to control the timing of SMC loading. For the replisome, we used a *dnaB*(ts) allele to control the timing of replication initiation. We asked whether we could pause replisomes at specific locations to better manage the collision site. HPUra is a specific inhibitor of DNA polymerase PolC in *B. subtilis*, which has been shown to stall replication but retain the replisome at the replication fork[40–42]. To test the effectiveness of HPUra in our experimental system, we combined HPUra treatment with temperature shift experiments using a *dnaB*(ts) strain. We monitored replication progression using MFA and examined replisome integrity using fluorescence microscopy of DnaX-YFP[35] (Fig. 3).

Cells were synchronized as described above (Fig. 3a, timeline). At 15 min after replication initiation, the cell cultures were treated with or without HPUra. In the absence of HPUra, replication progressed as expected (Fig. 3a, MFA plots). In the culture treated with HPUra, the DNA content remained unchanged over time (Fig. 3a, bottom panel

+HPUra). These results indicate that HPUra stalled replication nearly immediately, and the DNA remained intact without degradation.

To test whether the replisome remains bound to the replication forks after HPUra treatment, we visualized the DnaX-YFP foci using fluorescence microscopy (Fig. 3a, micrographs). Since our cells were sensitive to temperature but microscopy sample preparation and visualization were performed at room temperature, we attempted to fix the cells before visualization. However, all the fixation procedures we tried generated visualization artefacts that affected the quantification of replisome foci numbers. Thus, we performed these experiments in live cells without fixation. In the exponential growth phase, DnaX-YFP foci were detected in 99.4% of nucleoids, indicating active replication. After G1 arrest, we observed that 46.2% of nucleoids contained replisome foci, much higher than the 18% of cells with pre-existing replisomes as indicated by MFA (Supplementary Fig. 1b). The higher percentage was derived from a portion of *dnaB*(ts) cells that had initiated replication during sample preparation and imaging at room temperature. At 15 min after initiation, we observed replisome

foci in 91.7% of nucleoids, consistent with our earlier estimation by MFA results that ~95% of cells initiated replication (Fig. 1c and Supplementary Fig. 1). At 45 min, the percentage of nucleoids containing replisome foci dropped slightly due to the cells completing replication. In the presence of HPUra, the percentage of cells containing replisome foci remained largely unchanged over time (Fig. 3a, b). These results indicate that the addition of HPUra stalled replication effectively and replisomes remained at the replication forks, consistent with previous reports[41,42].

To test whether HPUra has an off-target effect on SMC translocation, we performed Hi-C on *dnaB*(ts) cells that were constantly arrested in G1 (Supplementary Fig. 4a). At 15 min after ParB induction (+IPTG 15 min), ~730 kb of DNA from each of the left and right arms was zipped by SMC. After another 15 min, with or without HPUra, DNA zipping proceeded to the same distance (Supplementary Fig. 4a). Therefore, in the absence of active DNA replication, HPUra does not affect SMC translocation; the effect of HPUra on SMC movement reported in the following sections is due to HPUra's effect on the replisome.

### Head-to-tail collisions between SMC and the replisome (two-sided)

In *B. subtilis*, SMC translocation is faster than DNA replication: in exponentially growing cells, SMC translocation is at ~50 kb/min while the replisome progresses at ~40 kb/min[29,43]. In our current experimental setup at 42 °C, we measured that SMC translocates at ~71 kb/min (Supplementary Fig. 4b) while the replisome progression is estimated to be at ~66 kb/min (Supplementary Fig. 1b). Thus, SMC that loads behind the replisome may catch up and collide with the replisome in a head-to-tail orientation. Below, we set up two experiments to examine this type of collision.

Many bacteria have *parS* sites that are very close to the replication origin[32]. *B. subtilis* has five origin-proximal *parS* sites at −1°, −3°, −4°, −5°, and +4°, respectively (Fig. 1a). After replisomes travel past these sites, SMCs loaded at these sites will track behind both replisomes and may catch up, colliding with both at about the same time. To understand the consequences of such two-sided head-to-tail collisions, we used a strain containing a single *parS* at the endogenous −1° site and a *dnaB*(ts) allele to allow for synchronization of DNA replication. In a control condition in which the replisome was synchronized but far ahead of the SMC, there was little chance of SMC-replisome collision, and the chromosome arms were gradually zipped at ~71 kb per minute (Fig. 4a, Supplementary Fig. 4b, Supplementary Movie 2). To set up an experiment for SMC-replisome collision, we allowed replication to initiate synchronously for 15 min, then added HPUra to pause replication forks (Fig. 4b, MFA plots). At the same time, we induced expression of ParB to load SMC, then monitored DNA zipping over time (Fig. 4b, Hi-C plots, Supplementary Movie 2). Compared to the control, the secondary diagonal in the collision experiment had three major features: (1) The Hi-C intensity had a sharp drop beyond the replisomes, suggesting that the majority of SMCs did not bypass the replisomes, but were either blocked or removed by the replisomes. (2) Although the intensity was faint, the secondary diagonal extended beyond the replisomes, suggesting that some SMCs were able to bypass the replisomes and continue zipping up the arms. Consistent with this idea, ChIP-seq experiments showed that SMC enrichment progressed beyond the replisomes to the same end positions of Hi-C zipping (Fig. 4c). Since the replisomes did not fall apart in HPUra-treated cells (Fig. 3), we think these Hi-C and ChIP-seq signals were from SMCs bypassing the stalled replisomes rather than SMCs unimpeded due to fork collapse. (3) The end positions of DNA zipping in the collision condition (Fig. 4b) lagged behind that of the control (Fig. 4a), indicating that the SMCs that bypassed the replisomes were delayed by the replisomes before bypassing. To estimate this delay, we measured that at 15 min after IPTG addition, with the replisomes, the DNA zipping distance on each replication arm was ~183 kb shorter than that of the

control (Supplementary Fig. 5). Using the measured extrusion rate of ~71 kb per minute, we inferred that a small fraction of SMCs paused for about three minutes before bypassing the replisomes (Supplementary Fig. 5a).

### Head-to-tail collisions between SMC and the replisome (one-sided)

Many bacterial genomes contain *parS* sites far away from the replication origin[32]. *B. subtilis* has four origin-distal *parS* sites at +17°, +42°, +91°, and −27°, respectively (Fig. 1a). After a replisome passes these sites, SMCs loaded behind the replisomes may catch up with only one replisome, generating a one-sided head-to-tail collision. To investigate this situation, we engineered a single *parS* site at −59° site (Fig. 5a, b). In the control experiment, SMC loading was induced in cells arrested in G1 without replication (Fig. 5a, MFA plots). The time-course Hi-C experiment showed that DNA zipping proceeded at ~71 kb per minute toward the terminus and ~49 kb per minute toward the origin, and the zipping was completed after 25 min of the induction (Fig. 5a, Supplementary Fig. 5b, Supplementary Movie 3). The asymmetry in SMC translocation speed toward the terminus versus toward the origin is consistent with previous observations and is likely caused by the arrangement of highly transcribed genes on the chromosome. Namely, in *B. subtilis*, the most highly transcribed genes are codirectional with DNA replication. For these genes, RNA polymerases in high density move in the same direction as the replisome. In the absence of DNA replication, SMCs translocating from the terminus to the origin still move against these dense RNA polymerases and might be slowed down by them[29,37,39,44].

To set up a one-sided head-to-tail collision in our experiment, we initiated replication and allowed the replisome to progress beyond the *parS* site (Fig. 5b, MFA plots). We added HPUra to pause replication and added IPTG to induce the SMC loading (Fig. 5b). In the time-course Hi-C experiment, compared with the control experiment without replisome present (Fig. 5a, Supplementary Movie 3), the secondary diagonal in the collision experiment had the same major features observed in the two-sided head-to-tail collision (Fig. 4b): DNA zipping extended beyond the collision site, but the intensity was much lower than that in the control, suggesting that the majority of SMCs were removed or paused by the replisome, and some SMCs bypassed the replisome. Moreover, the end position of DNA zipping lagged behind that of the control, indicating that the SMCs that bypassed the replisome had experienced some delay before bypassing. At 15 min after IPTG addition, we measured that with or without replisomes, the difference in zipping distance was ~129 kb toward the terminus and ~72 kb toward the origin (Supplementary Fig. 5b). Using the terminus-directed extrusion rate of ~71 kb/min and origin-directed extrusion rate of ~49 kb/min, we inferred that bypassing happened after a ~two-minute delay in this one-sided head-to-tail collision (Supplementary Fig. 5b). Consistent with the Hi-C results, ChIP-seq analyses revealed that SMC enrichment decreased beyond the collision site compared with the control (Supplementary Fig. 5c). Importantly, SMC enrichment not only decreased on left side of the *parS*, but also decreased on the right side (Supplementary Fig. 5c). This effect is similar to SMC being unloaded by a sequence-specific unloader, XerD[38].

### Head-on collisions between SMC and the replisome

Since all the *parS* sites have some distance from the replication origin, as the replisome travels from the origin toward the terminus, SMC loaded at the *parS* will collide with the replisome head-on. To explore the consequence of head-on SMC-replisome collisions in a simple experimental setup, we used the same strain containing a single *parS* site at −59° but changed the timing of ParB induction and replisome stalling. We induced SMC loading at the onset of replication initiation and added HPUra at 15 min after replication initiation, which was before the replisome encountered SMC (Fig. 5c). We found that

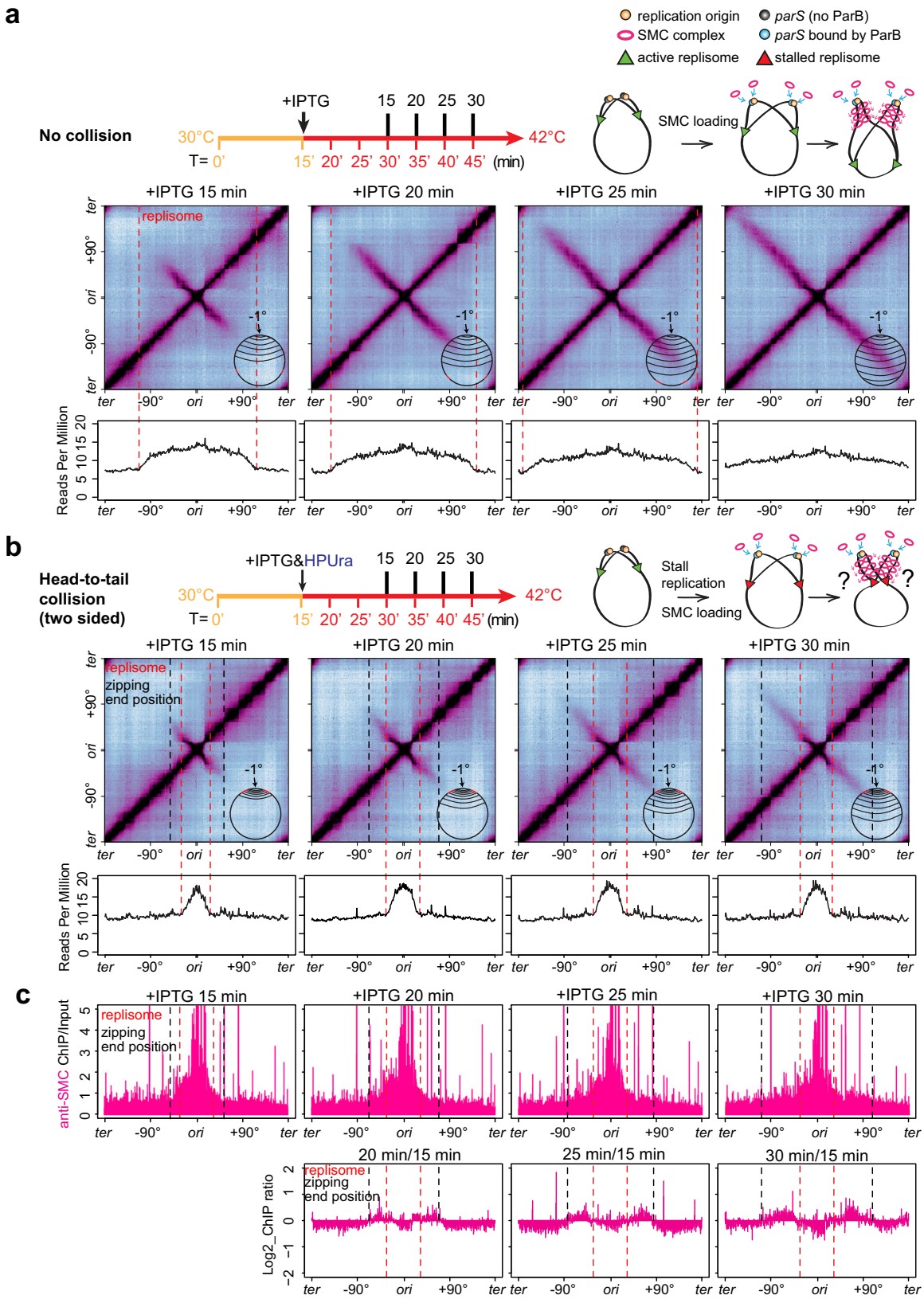

although the end position of DNA zipping lagged behind that of the control at each time point, the DNAs were zipped beyond the replisome, indicating that some SMCs bypassed the replisome. However, the intensity beyond the collision point was lower than that of the control in Fig. 5a, indicating that some SMCs were unloaded or paused at the collision site.

## Framework of simulations

Next, we used simulations to understand what rules of engagement between SMC and the replisome generated the observed Hi-C and ChIP-seq results. As described earlier, we first simulated replisome dynamics to reproduce the experimental replication profiles (Supplementary Fig. 1, also see details in Supplementary Methods). For SMC

**Fig. 4 | Two-sided head-to-tail collisions between SMC and stalled replisomes.** See also Supplementary Figs. 4 and 5 and Supplementary Movie 2. A two-sided head-to-tail collision between SMC and the replisome was set up by controlling SMC loading (using IPTG-inducible *parB*) and replication stalling (using HPUra) in cells synchronized for DNA replication (using a *dnaB*(ts) allele). The timeline for each experiment is shown on the top panels. Numbers below the timeline indicate synchronization status, in which T = 0′ is the onset of replication initiation; numbers above the timeline indicate the time after IPTG or HPUra addition as indicated. Schematics of experimental setups are shown in the top right panel. **a** A control experiment with no SMC-replisome collisions, in which the replisome is far ahead of SMC. A *dnaB*(ts) strain containing IPTG-inducible *parB* and a single *parS* at −1° (BWX4310) was grown as indicated by the timeline (top panel). Hi-C contact maps were generated from cells at 15 min, 20 min, 25 min, and 30 min after IPTG addition. Schematics of the zipped region are superimposed on the Hi-C maps (middle

panel). MFA plots from the same samples in Hi-C experiments are shown (bottom panel). The positions of replication forks are indicated with red dashed lines. **b** A two-sided head-to-tail collision in which both SMC motors encounter stalled replisomes. In the same strain shown in (**a**), HPUra was added to stall replication at 15 min after replication initiation. At the same time, IPTG was added to induce SMC loading. The positions of replication forks and the end positions of DNA zipping are indicated with red dashed lines and black dashed lines, respectively. **c** anti-SMC ChIP-seq was performed on the same samples as in (**b**). ChIP enrichment (ChIP/input) was plotted in 1-kb bins (top panel). The ratio of ChIP enrichments at indicated time points relative to +IPTG 15 min was plotted in $\log_2$ scale in 5-kb bins to track SMC translocation beyond the stalled replisomes (bottom panel). The positions of replication forks and the end positions of DNA zipping are indicated with red dashed lines and black dashed lines, respectively.

dynamics, we adapted a framework based on previous measurements and simulations[37,39,45] and included parameters specific to this study (see details in Supplementary Methods). Briefly, we set the number of SMC complexes (loop-extruding unit) to be 40 per chromosome. We considered that one loop-extruding unit contained two independent extrusion motors translocating away from the loading site. If translocation by one motor was blocked by an encounter with the replisome, the other continued its extrusion[29,37,39]. SMC complexes were allowed to load anywhere on the chromosome, with a loading preference at the *parS* site, so that they most often loaded at the *parS*. In addition, we determined two parameters that were specific for this study's experimental conditions: by simulating SMC distribution to match the experimental ChIP-seq result, we estimated that SMC's spontaneous disassociation rate at 42 °C was about 1200 s (Supplementary Fig. 6, also see details in Supplementary Methods); by directly measuring experimental Hi-C results, we determined that the SMC translocation rates at 42 °C was ~71.5 ± 2 kb/min per motor (i.e., for a total rate of loop growth of ~143 ± 4 kb/min) (Supplementary Fig. 4b).

When SMC reached a replisome, we first considered three outcomes[20] in our simulations: blocking, in which only the collided SMC motor was stopped by the replisome[29,37,39]; unloading, in which both motors of the whole SMC complex were removed by the replisome[38]; and bypassing, in which SMC traversed the replisome and continued translocating. Given the indications from experiments, we next extended these simple models by allowing SMC to pause in a blocked state for a varying amount of time before unloading or bypassing. We simulated SMC dynamics and calculated SMC occupancy on the chromosome to compare with the experimental ChIP-seq results. Using the chromosomal positions of SMCs, we generated simulated contact frequency maps to compare with experimental Hi-C results (see details in Supplementary Methods).

### Determining the rules for head-to-tail SMC-replisome collisions
We simulated the rules of engagement for SMC and the replisome in the head-to-tail collision shown in Fig. 5b. We used the experimental data at 25 min following IPTG treatment (Fig. 6a) and first explored the three basic rules of engagement: bypassing-only, blocking-only, and unloading-only (Fig. 6b–d). In each model, the simulated SMC ChIP enrichment profiles (Fig. 6b–d, orange lines in top panels) and Hi-C maps (Fig. 6b–d, bottom panels) displayed specific features that were different from the experimental data (Fig. 6a).

In the bypassing-only model, the simulated SMC enrichment beyond the collision site was much higher than the experimental ChIP result (Fig. 6b, top panel). The simulated Hi-C map had the DNA regions flanking the *parS* site fully zipped, with intensities very similar to the control which had no collision between SMC and the replisome (the simulated Hi-C map in Fig. 6b compared with the experimental Hi-C map in Fig. 5a). These results indicate that the bypassing-only model allowed too many SMCs to move beyond the replisome.

In the blocking-only model, simulated SMC occupancy showed a sharp peak at the collision site (near −90°), which was absent from experimental ChIP-seq data. Additionally, SMC enrichment on the right arm was much higher than the experimental data (Fig. 6c, top panel). The simulated Hi-C map showed an intense "square" of interactions at the center of the map (Fig. 6c, bottom panel), which was absent from the experimental Hi-C map (Fig. 6a, bottom panel). Therefore, we ruled out the blocking-only model.

The unloading-only model exhibited a better match to the experimental data, but the ChIP signal near the loading site was higher than the experimental data (Fig. 6d, top panel). The simulated Hi-C interaction frequency beyond the collision site diminished much faster than the experimental Hi-C data (Fig. 6d, bottom panel).

Our simulation results indicate that the single rule of engagement (bypassing, blocking, or unloading) did not fully reproduce the experimental data. We next explored a combination of these engagement modes in the simulation to achieve a better fit to the experimental data (Supplementary Fig. 7). We assumed that when SMC collided with a replisome, the collided SMC motor was blocked, then it either bypassed the replisome or unloaded from the chromosome at specific rates (see details in Supplementary Methods). We generated simulated SMC occupancy profiles across a broad range of bypassing and unloading times to compare with the experimental data (Supplementary Fig. 7a). Intuitively, when the bypassing time was short and the unloading time was long, the simulated result resembled the bypassing-only model (Fig. 6b, Supplementary Fig. 7a); when both the bypassing and unloading times were long, the result resembled the blocking-only model (Fig. 6c, Supplementary Fig. 7a); when the unloading time was short and the bypassing time was long, the result resembled the unloading-only model (Fig. 6d, Supplementary Fig. 7a). To find the optimal model matching the experimental data, we performed the goodness-of-fit analysis (Supplementary Fig. 7b, also see details in Supplementary Methods) and visual inspections of the overall shape of SMC enrichment profiles to assess the similarity between the simulations and the experiment. We found that if the unloading was more than two-fold faster than the bypassing, the simulation produced a better fit than otherwise (Supplementary Fig. 7b, 1:2 ratio line). A few combinations produced a very good fit (Supplementary Fig. 7b, black asterisks). Since the experimental Hi-C analysis estimated the bypassing time to be about two minutes for this one-sided head-to-tail collision (Supplementary Fig. 5b), the best combination in the range was unloading at 30 s and bypassing at 120 s (Supplementary Fig. 7b, black check mark in the white rectangle), which produced simulated ChIP and Hi-C results better matching the experimental data than models involving the single rule of engagement (Fig. 6e compared with Fig. 6b–d).

### Determining the rules of SMC-replisome head-on collisions
Next, we investigated the rules of engagement for SMC and the replisome in the head-on collision shown in Fig. 5c. We simulated SMC occupancy profiles and Hi-C results at 25 min after IPTG induction

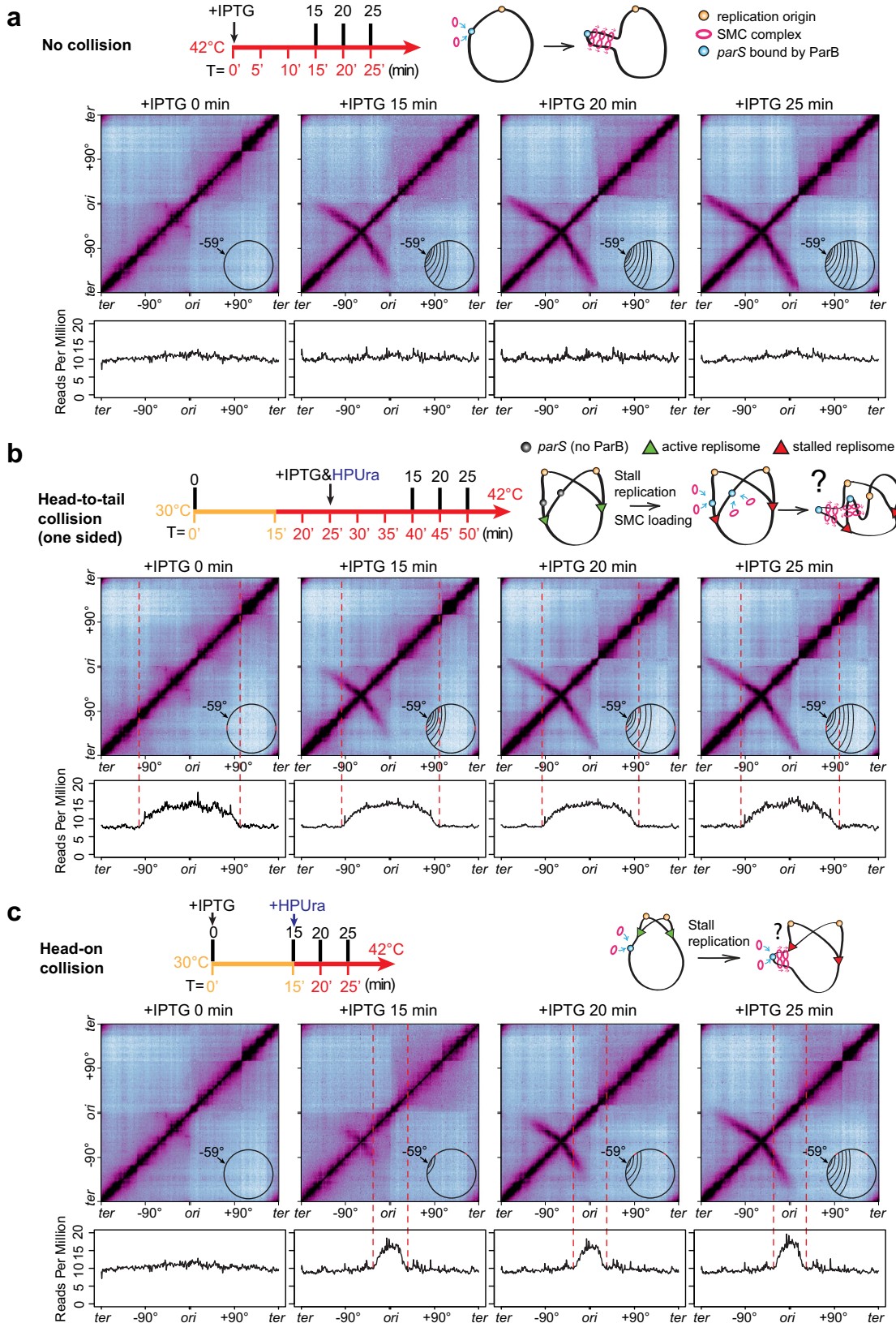

(Fig. 6f–j). In a trend similar to the simulations of the head-to-tail collision, simple models of bypassing, blocking, or unloading did not reproduce the experimental results (Fig. 6f–i). Next, we swept a wide range of unloading and bypassing times (Supplementary Fig. 8a). The goodness-of-fit analysis and visual inspections identified four combinations that produced much better fits than each of the simple models

(Supplementary Fig. 8b, black asterisks). Among these combinations was the same rule of engagement seen for the head-to-tail collision (unloading = 30 s and bypassing = 120 s) (Fig. 6j, Supplementary Fig. 8b, black check mark).

Although our simulations did not narrow the unloading and bypassing times to one specific set of numbers, we found that

**Fig. 5 | One-sided head-to-tail and head-on collisions between SMC and stalled replisomes.** See also Supplementary Fig. 5 and Supplementary Movie 3. **a** A control experiment with no SMC-replisome collision using a *dnaB*(ts) strain containing IPTG-inducible *parB* and a single *parS* at −59° (BWX5297). This control was set up by growing G1-arrested cells at 42 °C constantly (see the timeline in the top panel). Hi-C contact maps (middle panel) and the MFA plots (bottom panel) are shown for samples at 0 min, 15 min, 20 min, and 25 min after IPTG addition. **b** A one-sided head-to-tail collision in which one motor of the SMC complex encounters a stalled replisome. In the same strain shown in (**a**), after the leftward replication fork moved

beyond the *parS* site, the replisomes were stalled by the addition of HPUra. At the same time, SMC loading was induced by IPTG. Hi-C contact maps (middle panel) and the MFA plots (bottom panel) are shown for samples at 0 min, 15 min, 20 min, and 25 min after IPTG addition. The positions of replication forks are indicated with red dashed lines. **c** A head-on collision between SMC and a stalled replisome. In the same strain shown in (**a**), the replisomes were stalled before colliding with SMC. Hi-C contact maps (middle panel) and the MFA plots (bottom panel) are shown from samples at 0 min, 15 min, 20 min, and 25 min after IPTG addition. The positions of replication forks are indicated with red dashed lines.

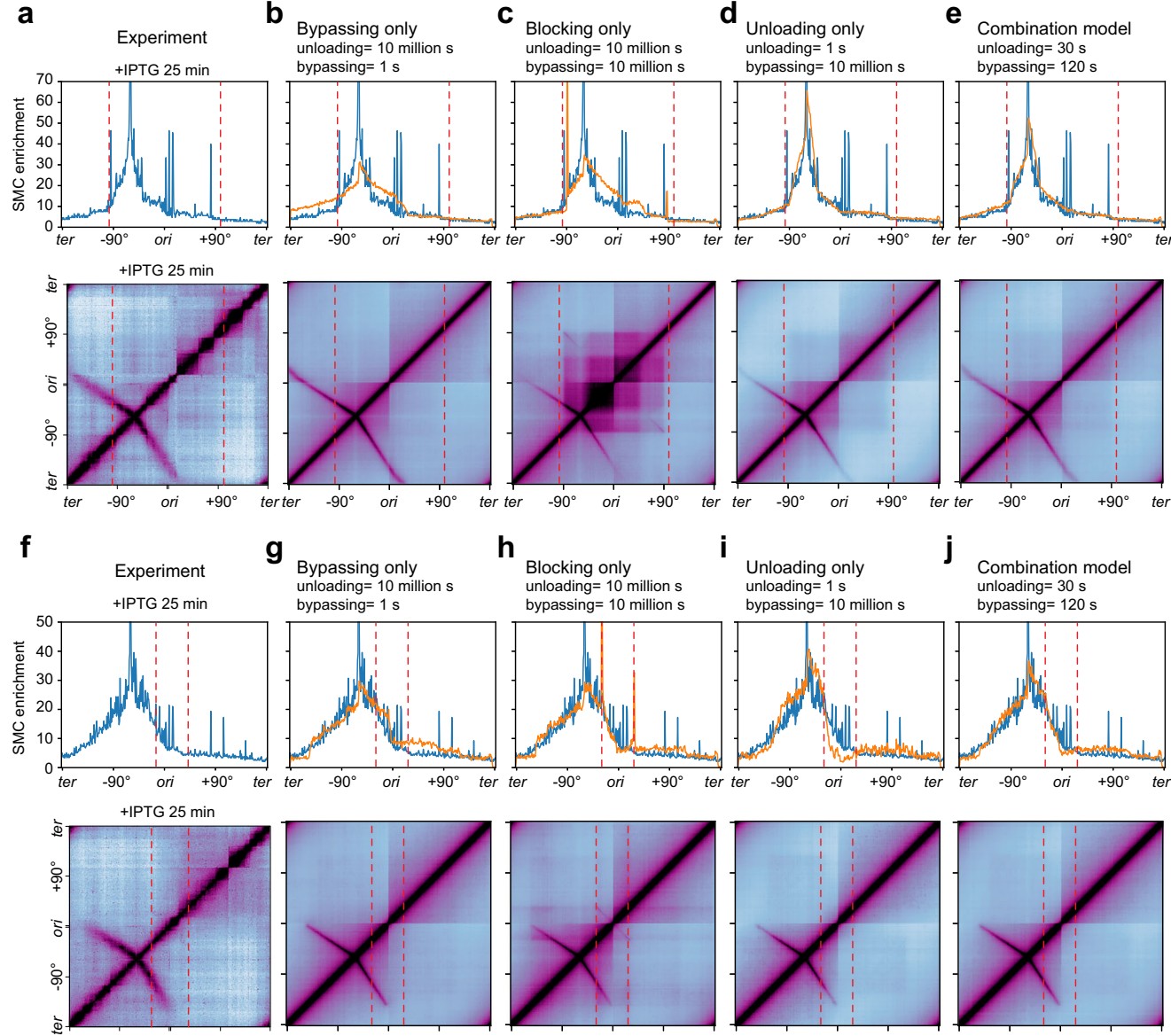

**Fig. 6 | Simulation results for head-to-tail and head-on SMC-replisome collisions.** See also Supplementary Figs. 6, 7, and 8. **a** Experimental data for the head-to-tail collision shown in Fig. 5b (+IPTG 25 min). Top: anti-SMC ChIP-seq was filtered as described in Supplementary Methods and plotted as reads per million in 10-kb bins. Bottom: Hi-C contact map. The positions of replication forks are indicated with red dashed lines. **b**–**e** Top: simulated SMC distributions (orange curves) plotted on top of experimental ChIP-seq result (blue curves). Bottom: simulated Hi-C maps. Four models are shown respectively for bypassing-only (**b**), blocking-only (**c**), unloading-only (**d**), and the combination model (**e**). The complete parameter sweep is shown in Sup-

plementary Fig. 7. (**f**) Experimental data for the head-on collision shown in Fig. 5c (+IPTG 25 min). Top: anti-SMC ChIP-seq was filtered as described in Supplementary Methods and plotted as reads per million in 10-kb bins. Bottom: Hi-C contact map. The positions of replication forks are indicated with red dashed lines. (**g**–**j**) Top: simulated SMC distributions (orange curves) plotted on top of experimental ChIP-seq result (blue curves). Bottom: simulated Hi-C maps. Four models are shown respectively for bypassing-only (**g**), blocking-only (**h**), unloading-only (**i**), and the combination model (**j**). The complete parameter sweep is shown in Supplementary Fig. 8.

unloading is the major rule, and bypassing is less frequent. Although it is possible that SMC-replisome head-on collisions and head-to-tail collisions use different unloading and bypassing rates, we found certain combinations, such as unloading at 30 s and bypassing at 120 s, can explain both types of collisions. This set of numbers signifies that, when SMC encounters the replisome, on average, it first pauses for ~24 s which is much shorter than the spontaneous dissociate rate of ~1200 s we estimated (Supplementary Fig. 6b). Then with a ~80% chance, SMC unloads from the chromosome, and with a ~20% chance, SMC bypasses the replisome and continues translocation.

## SMC colliding with a moving replisome

Previous experiments (Figs. 4 and 5) and simulations (Fig. 6) used stalled replisomes. Although this simplified the collision scenarios, it did not account for replisome movement and activity. Our earlier experimental results show that the replisome moves unhindered when it collides with SMCs (Fig. 1c). We therefore hypothesized that when a SMC is blocked by a moving replisome, head-on collisions will result in the replisome pushing the SMC; while in head-to-tail collisions, the replisome will slow the SMC to match the speed of the replisome. To test this hypothesis and determine if the rules of engagement established using stalled replisomes apply to unperturbed, moving replisomes, we conducted two SMC-replisome collision experiments with active replisomes (Supplementary Movie 4).

In the first experiment, using the same strain as in Fig. 5, we induced SMC loading upon replication initiation and allowed replisomes to progress through the cell cycle (Fig. 7a). MFA and Hi-C results at T = 15–20 min showed SMCs colliding head-on with the moving replisome. Compared with experiments containing stalled replisomes, collisions with moving replisomes exhibited more pronounced features (Supplementary Movie 4). Specifically, as indicated by the Hi-C maps, DNA zipping progressed more slowly, and the secondary diagonal displayed a stronger downward tilt (Supplementary Fig. 9a, b); on ChIP-seq plots, SMC enrichment had a slower progression and stronger buildup at the collision zone (Supplementary Fig. 9c, d, black arrows), although SMC enrichment on the terminus-proximal side remained unaffected. These experimental results support a model in which the SMC motor in a head-on collision is blocked and pushed back by the moving replisome, while the unblocked SMC motor continues translocating. Despite the blocking and pushback, DNA zipping eventually extended beyond the replisome, albeit at lower Hi-C intensity, indicating that some SMCs bypassed the moving replisome after the head-on collision. Following these head-on collision events, the intensity of the secondary diagonal was reduced (compare 25 min with 20 min in Fig. 7a; compare the moving replisomes with the stalled replisomes in Supplementary Fig. 9a, b, 25 min), suggesting that some SMCs were unloaded from the chromosome upon the head-on collisions. After the replisome passed the −59° parS site (Fig. 7a, T = 25–40 min), SMCs loaded at the parS trailed the replisome in a head-to-tail orientation. The upward tilt of arc 2 on the Hi-C maps (Fig. 7a, T = 25–40 min, black arrows) indicates that SMCs were slowed down behind the moving replisome. Arc 2 lagged considerably farther behind the leftward replisome, suggesting that SMCs did not catch up with or bypass the replisome under these conditions. Therefore, in a head-on collision, the rules of engagement established with stalled replisomes appear to apply to the case of active replisomes; in a head-to-tail collision, additional rules appear to be at play, leading to the formation of arc 2.

In a second experiment, we used a strain with continuous SMC loading at the −59° parS site (Fig. 7b, Supplementary Movie 4). This strategy allowed SMC to pre-load on the chromosome, thereby increasing the frequency of SMC-replisome collisions, particularly head-on encounters (T = 0–20 min). Hi-C analyses revealed evidence of blocking, bypassing, and unloading (Supplementary Movie 4). First,

the secondary diagonal showed a pronounced downward curve at 10–15 min (Fig. 7b, compare with 0 min, yellow carets), indicating that the moving replisome blocked and pushed SMCs in the head-on orientation (Fig. 7b, schematics 0 min and 15 min). Second, from 10 to 20 min, despite the replisome traversing the region between the origin and the parS, this left arm region remained zipped with the terminus-proximal region (Fig. 7b, schematic 20 min). This persistent interaction indicates that some SMCs bypassed the replisome after the head-on collisions, as blocking and unloading alone would preclude such zipping (Fig. 7b, schematic inset). The fainter secondary diagonal generated by these bypassing SMCs (Fig. 7b, compare 10–25 min with 0 min after replication initiation) suggests that a fraction of SMCs was unloaded upon the collisions. Finally, arc 2 also appeared, consistent with the observations in Fig. 7a.

Collectively, our experiments demonstrate that SMC complexes, upon colliding with moving replisomes, undergo momentary blocking followed by either unloading from the DNA or bypassing the replisome. This set of engagement rules can also explain the results in our earlier experiment in Fig. 2a, where we used a strain containing a −27° parS with continuous loading of SMC and moving replisomes. Finally, it is noteworthy that in the three different time-course experiments with the moving replisomes (Figs. 2, 7a, b), the replication profiles were similar regardless of the parS location and the timing of SMC loading, reinforcing the notion that SMC translocation does not affect replisome progression.

## Simulations to understand the formation of arc 2 on the Hi-C map

In the three independent experiments with moving replisomes discussed above (Figs. 2, 7a, b), we observed the formation of arc 2, a feature absent when SMC collided with stalled replisomes (Fig. 5). Arc 2 only formed after the leftward replisome passed the parS site. This suggests that arc 2 was generated by SMCs loaded at the parS, which then chased the replisome toward the terminus in a head-to-tail manner (Fig. 7b, schematic of 30–40 min after replication initiation). Generally, the angle of SMC zipping reflects the relative speed of SMC movement on the two DNA arms flanking the loading site (Supplementary Fig. 10a–d)[29,37]. For example, a secondary diagonal perpendicular to the primary diagonal (Supplementary Fig. 10a, −1° parS) indicates similar speeds of SMC movement on both chromosome arms. A downward tilt (Supplementary Fig. 10b, −59° parS) indicates faster SMC movement toward the terminus than toward the origin. Conversely, an upward tilt (Supplementary Fig. 10c, d) indicates slower SMC movement toward the terminus than toward the origin. Measurements from experimental Hi-C data indicate that on arc 2, the SMC motor chasing the replisome toward the terminus moves at ~0.2× speed of the SMC motor moving toward the origin (Supplementary Fig. 10c, d). Given that SMC translocates toward the origin at ~49 kb/min (Supplementary Fig. 5b), the SMC motor chasing behind the replisome slowed to an average speed of ~10 kb/min. This is ~0.14× of SMC's standard translocation speed toward the terminus in a situation where SMC is unhindered by replisomes (~71 kb/min). Since the estimated replisome speed is ~66 kb/min (Supplementary Fig. 1b), our experimental results indicate that SMCs tracking behind the moving replisome translocated at a speed much slower than the replisome itself. Thus, the slowdown cannot be explained solely by SMCs being blocked by the replisome itself, because that would retard SMC speed to match the replisome speed (~66 kb/min) and result in SMCs tracking immediately behind the replisome. Consistent with this, the end positions of arc 2 were distant from the replisomes at all time points (Supplementary Fig. 10e, f). One possibility is that other factors associated with the moving replisome impede SMCs. For example, precatenanes, which form behind the replication fork and take ~15 min to be resolved by topoisomerases[46,47] may create a topological barrier for SMC passage. If this is the case, the slowdown would be temporary and

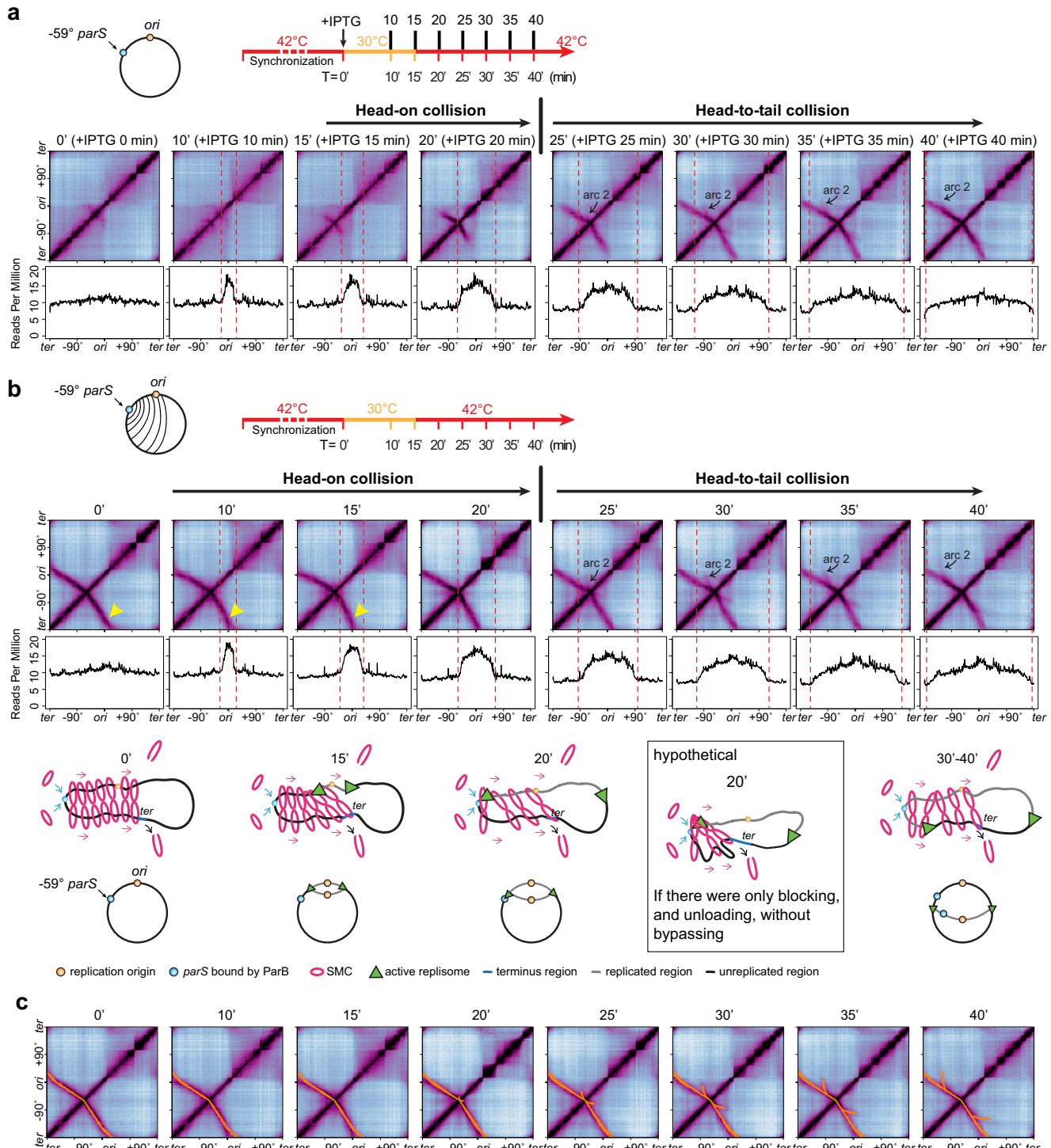

**Fig. 7 | The collisions between SMCs and moving replisomes.** See also Supplementary Figs. 9–11 and Supplementary Movie 4. **a** Hi-C maps and MFA plots generated from a time-course experiment using a *dnaB*(ts) strain containing IPTG-inducible *parB* and a single *parS* at −59° (BWX5297), which is the same strain used in Fig. 5. SMC loading was induced at the onset of replication initiation. Here, replisomes were allowed to progress through the cell cycle, in contrast to Fig. 5 in which replisomes were stalled using HPUra. **b** Hi-C maps and MFA plots generated from a time-course experiment using a *dnaB*(ts) strain containing a single *parS* at −59°

*parS* (BWX5529). In this strain, SMC was loaded constantly, in contrast to IPTG-induced SMC loading seen in (**a**). Schematics depicting SMC translocation at indicated time points are shown in the lower panel. **c** Simulations of SMC colliding with a moving replisome to reproduce Hi-C maps shown in (**b**). At the indicated time points, the final position of each SMC complex was simulated and plotted as an orange dot. Because of the high number of sampled SMC complexes, the dots are not individually resolvable but form lines. A fork memory time of 700 s was used. See Supplementary Fig. 11 and Supplementary Methods for details.

confined to the region immediately behind the replication fork, creating a transient "memory effect" on the recently traversed DNA.

To investigate how transient alterations in SMC speed within regions recently traversed by a replication fork could generate arc 2,

we developed a specialized, stand-alone simulation. This model tracked pairs of SMC motors and a moving replication fork, focusing on the concept of "fork memory" – a lingering effect of the fork's passage that temporarily modified the rate of the affected SMC motor.

To specifically isolate this effect, the model simplified certain interactions. Namely, we did not include SMC-SMC interactions, or SMC being paused and pushed by the replisome (see Supplementary Methods). Instead, inspired by factors like precatenanes and based on the hypothesis that SMC's speed is progressively reduced when it translocates through a region recently passed by a replication fork, our model incorporated this slowdown (conceptualized in Supplementary Fig. 11a): far from the replisome, SMC's speed is unaffected (i.e., SMC retains its standard speed); upon an SMC motor entering the "fork memory zone", the motor's speed decreases with decreasing distance to the fork; when the SMC motor reaches the replisome's current position from behind, the motor's speed falls to zero. Our model captured this behavior using a parameter defining the distance of this memory zone behind the fork, the "fork memory distance", or equivalently, the "fork memory duration". Other than this new parameter, the simulation included parameters that were used in our earlier simulations, such as the speed of the replisome, the stochastic DNA polymerase loading time at replication initiation, SMC's spontaneous dissociation rate, and SMC's loading rate at the *parS* versus elsewhere (see Supplementary Methods).

Using experimental Hi-C data as the basis (Supplementary Fig. 11b), we performed parameter sweeps. These sweeps varied two simulation parameters, the "fork memory duration" and the effective simulation time, to reproduce the observed positions and angles of arc 2 (see Supplementary Methods). We found that a memory duration of approximately $700 \pm 100$ s (10–13 min) provided the best fit to the experimental data in these examples (Supplementary Fig. 11c, d). This duration corresponds to a conceptual "memory distance" of 660–880 kb, assuming a fork speed of ~66 kb/min as obtained experimentally. A "fork memory" in this range (e.g., ~700 s of duration, or ~770 kb in distance), reproduced the arc 2 feature across all relevant time points in the time-course Hi-C experiments (Fig. 7c and Supplementary Fig. 11e). To validate these simulation-derived parameters, particularly the fork memory distance, we experimentally identified the position where SMCs began to slow down (i.e., where arc 2 diverged from line 2; Supplementary Fig. 10e, f). Our measurements indicate that SMCs started slowing ~605–705 kb behind the replisome, corresponding to ~9–10 min of SMC translocation at the unaffected speed (~71 kb/min) (Supplementary Fig. 10e, f). This experimental estimation of SMC slowdown is largely consistent with the simulation-derived optimal fork memory (~660–880 kb in distance or ~10–13 min in time). Our experiments and simulations suggest that SMCs catching up with a moving replisome from behind indeed experience a significant slowdown for a duration of ~9–13 min, corresponding to a physical distance of a ~605–880 kb behind the active replication fork.

We note that arc 2 was absent from the head-to-tail collision experiments with stalled replisomes in Fig. 5b. This suggests that the unknown factor associated with moving replisomes, which slowed SMCs to generate arc 2, was resolved before SMCs reached the stalled replisome. To estimate this timeframe, we measured the time for SMCs to reach the stalled replisome using experimental MFA and Hi-C data. In Fig. 5b, before replisome stalling (by HPUra) and ParB induction (by IPTG), the leftward replisome had progressed ~1169 kb from the origin. This position is ~517 kb from the −59° *parS* site, which would take SMC ~7 min to traverse (517/71≈7 min). However, considering the ~5 min lag time for SMC loading after IPTG addition (Supplementary Fig. 4b), the total time for SMCs to reach the stalled replisome is ~12 min after IPTG addition. The absence of arc 2 in Fig. 5b implies that the slowdown factor must be resolved within ~12 min. This duration is consistent with our earlier estimates that the unknown factor slows SMCs for ~9–13 min behind the replisome.

## Discussion

SMC complexes are major chromosome organizers in all domains of life. They load on the DNA and extrude DNA loops up to millions of

base pairs in size, during which they encounter various DNA-bound molecules on the crowded chromatin fiber. Except for specific regulators, such as CTCF, which blocks cohesin movement[48–51], a cohesin interaction motif in the minichromosome maintenance (MCM) complexes that pauses cohesin[52], and the bacterial XerD protein, which unloads SMC at the terminus region[38], SMC can bypass many molecules, including nucleosomes[53], RNA polymerases[37,54], nucleoid-associated proteins[55], and other SMC molecules[39,56,57]. The ability for SMCs to bypass barriers not only promotes chromosome compaction and segregation, but also facilitates the trafficking of other factors along the chromosome[45,58,59]. One important molecular machine that SMCs frequently encounter is the replisome. The in vivo consequence of their collisions and how these collisions affect DNA replication and chromosome compaction were unknown. Here, taking advantage of well-controlled SMC loading and replisome progression in *B. subtilis*, we have determined the in vivo consequences of collisions between SMC and the replisome, and investigated the effects of SMC-replisome collisions on DNA replication and chromosome folding. Our experiments and simulations have demonstrated that the progression of DNA replication is unaffected by SMC-mediated loop extrusion. By contrast, replisomes restrict loop extrusion by blocking and then unloading SMC complexes (with a ~80% chance). However, occasionally (with a ~20% chance), SMC complexes can bypass the replisome and continue translocating.

### A head-on collision versus a head-to-tail collision

Our experiments with stalled replisomes (Figs. 4, 5, and Supplementary Movies 2, 3) indicate that for both head-on and head-to-tail SMC-replisome collisions, a combination of blocking, unloading, and bypassing must happen to generate the observed results (Fig. 8a–c). Our simulations show that unloading is more frequent than bypassing, and a combination of these modes of engagement (unloading = 30 s and bypassing = 120 s, on average) (Supplementary Figs. 7b, 8b) can reproduce the prominent features of the experimental results for both head-on and head-to-tail collisions. These matching parameters suggest that when SMC encounters a stalled replisome, regardless of the orientation, the replisome itself presents as an obstacle to SMC movement. In the case of moving replisomes, our experiments indicate that when SMC collides with a replisome in a head-on orientation, a combination of blocking, unloading, and bypassing must happen to generate the observed results (Figs. 2, 7, head-on collision). However, when it trails behind a moving replisome in a head-to-tail orientation, SMC is further slowed down and with little chance to reach or bypass the replisome (Figs. 2, 7, head-to-tail collision). Our experiments and simulations suggest that SMC is delayed for an average of 11 min (~9–13 min) behind the moving replisome (Supplementary Figs. 10e, f, 11), which is about the time needed to resolve precatenanes generated behind the replication forks[46,47] (Fig. 8d). However, the exact mechanism of this further slowdown is unknown. Behind the replication fork, there are single-stranded DNAs and their binding proteins, Okazaki fragments, precatenated sister chromatids, hemimethylated DNAs, and more. Any of these factors or a combination of them may contribute to the slowdown.

### SMC bypassing the replisome

Although the probability of SMC to bypass the replisome is low (~20% or less), it is clear that bypassing is happening, as evidenced by DNA zipping beyond the stalled replisome in both head-on and head-to-tail collisions (Figs. 4, 5 and Supplementary Movies 2, 3), and DNA zipping beyond the moving replisome in head-on collisions (Figs. 2, 7 and Supplementary Movies 1, 4). SMC bypassing the replisome could hinder chromosome segregation because this allows SMC to tether newly replicated DNA with unreplicated DNA (see black dashed lines noting "after bypassing" in Fig. 8b, c). Fortunately for cells, SMC loading is strongly biased to the *parS* sites that are close to the replication origin,

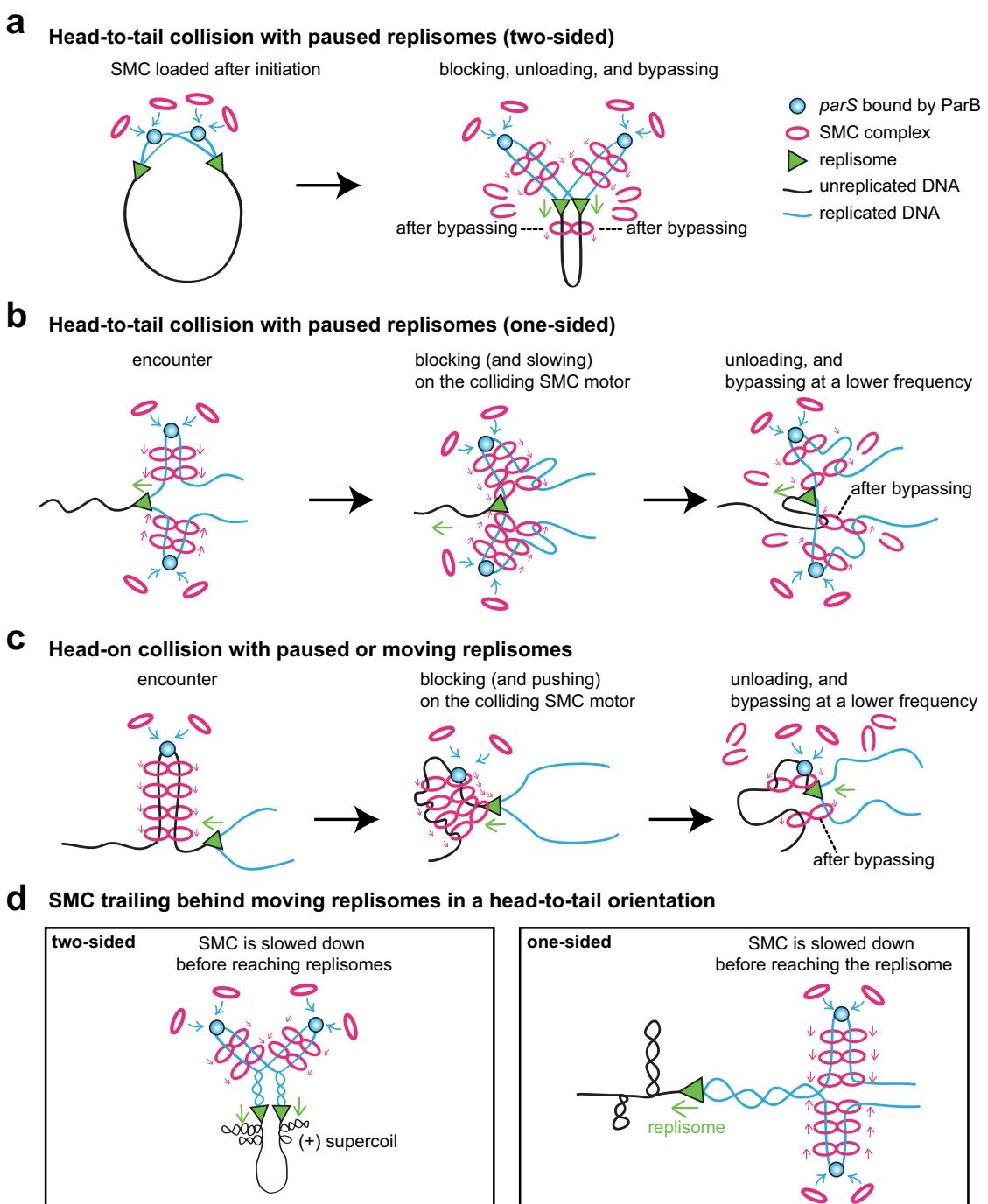

**Fig. 8 | Schematic models for the consequence of SMC-replisome collisions.** Models for SMC interacting with stalled replisomes in a two-sided head-to-tail collision (**a**), a one-sided head-to-tail collision (**b**), or a head-on collision (**c**). Models for SMC trailing behind moving replisomes in a head-to-tail orientation (**d**).

because *parS* sites are frequently very close to the replication origin[32], the most origin-proximal *parS* sites have higher binding affinity for ParB[60], and the origin region has the highest copy number in the chromosome (Fig. 1b, MFA). Moreover, the rates of movement of the SMC complex and the replisome are closely matched in different species: for example, in *B. subtilis* growing at 42 °C, the replisome moves at ~66 kb/min and the SMC complex moves at ~71 kb/min (determined in the study); in *Caulobacter crescentus*, the replisome moves at ~21 kb/min and the SMC complex moves at ~16-19 kb/min[61,62]. The matching translocation rates of the two machines and the origin-biased SMC loading indicate that in wild-type cells, the vast majority of SMC complexes are "chasing" the replisome in a head-to-tail

orientation (Fig. 8a). This combination should result in few cases of SMCs bypassing the replisome, and if they do so, it should still not result in unwanted chromosomal tethers or chromosome segregation issues (Fig. 8a, after bypassing). Additionally, our experiments and simulations suggest that SMCs chasing the replisome are slowed down even before reaching the replisome (Fig. 8d, Supplementary Figs. 10e, f, 11), further lowering the chance of SMCs bypassing the replisome in real life.

### SMC-replisome collisions in eukaryotes
In eukaryotes, SMC cohesins establish sister chromatid cohesion during DNA replication; thus, the engagement between cohesin and

the replisome has fundamental implications for DNA replication and segregation. Several recent in vitro single-molecule studies examined the outcomes of cohesin-replisome collisions. Using *Xenopus* egg extracts and a naked DNA substrate, it has been found that when colliding with the replisome head-on, cohesin is frequently blocked and pushed by the replisome to the converging point of replication forks. But at some frequency, cohesin unloads from the replisome, and at a lower frequency, cohesin bypasses the replisome[63]. A separate study using *Xenopus* egg extract and a recent report using budding yeast cohesin both showed cohesin-replisome collision outcomes similar to the study mentioned above, although at different frequencies of pushing, unloading, and bypassing[64,65]. In vivo, the MCM complex, which is a DNA translocase and a catalytically inactive DNA helicase until S phase, was shown to restrict cohesin-mediated DNA-loop extrusion and allow cohesin to bypass it with a ~80% chance[52]. However, these results were from G1-arrested cells, where MCM was inactive for DNA unwinding and did not represent the effect of an active replisome. Our in vivo findings for SMC-replisome collisions in bacteria mirror the in vitro observations for cohesin-replisome collisions[63–65], establishing that the single-molecule results reflect what is happening inside cells. Moreover, our study provides interaction rules between SMCs and active replisomes occurring in the presence of all the other DNA transactions in vivo, which have not been investigated previously.

### Replisome movement is unaffected by SMC

Although SMC movement is restricted by the replisome, our experiments show that regardless of the location and the timing of SMC loading, the progression of DNA replication is unaffected (Figs. 1c, 2a, 7a, 7b). Single-molecule experiments also showed that replisomes were rarely affected by cohesins[63,65]. These findings highlight the mechanical prowess of the replisomes to clear or bypass protein roadblocks[66,67], which is essential for DNA replication and genome integrity. A future challenge is to understand the molecular mechanism for SMC unloading and bypassing at the replisome, for instance, to identify the replisome component(s) responsible for SMC unloading, to understand whether the ring-shaped SMC complex opens when it bypasses the replisome, and to investigate the factor(s) that further slows down SMCs trailing behind the replication fork. Future biochemical, structural, and single-molecule experiments are needed to address these questions.

In summary, replisome-mediated DNA replication and SMC-mediated chromosome organization and segregation are essential for cell viability. The resolution of collisions between the replisome and SMC complexes ensures faithful chromosome replication and segregation. Our experiments show that SMC complexes exhibit blocking, unloading, and bypassing behaviors when encountering the replisome in vivo, similar to in vitro results obtained for eukaryotic cohesin-replisome engagements. This convergence suggests that the interplay between SMC complexes and the replisome may represent a conserved mechanism for coordinating DNA replication and chromosome organization across diverse organisms.

## Methods
### General methods

*B. subtilis* strains were derived from the prototrophic strain PY79[68]. Cells were grown in defined rich Casein Hydrolysate (CH) medium at 22 °C, 30 °C, or 42 °C as specified. ParB production was induced by the addition of 1 mM IPTG (Dot Scientific, DS102125). To arrest replisome movement, 162 μM HPUra[40–42] was used. Lists of strains, plasmids, and oligonucleotides can be found in Supplementary Table 1. The list of Next-Generation-Sequencing samples can be found in Supplementary Data 1. Plasmids and strains generated in this study are available from the corresponding author with a completed Materials Transfer Agreement.

### Hi-C

The Hi-C procedure was carried out as previously described[35,69]. Specifically, cells grown under the desired conditions were crosslinked with 3% formaldehyde (Sigma, F8775-500ML) at room temperature for 30 min, then quenched with 125 mM glycine (Fisher Scientific, 02-002-947). Cells were lysed using Ready-Lyse Lysozyme (Lucigen, R1802M) and treated with 0.5% SDS. Solubilized chromatin was digested with HindIII (NEB, R0104M) for two hours at 37 °C. The digested ends were filled in with Klenow (NEB, M0210L) and Biotin-14-dATP (Fisher Scientific, 19-524-016), dGTP (Fisher Scientific, 10-297-018), dCTP (Fisher Scientific, 10-297-018), and dTTP (Fisher Scientific, 10-297-018). The products were ligated with T4 DNA ligase (NEB, M0202M) at 16 °C for about 20 h. Crosslinks were reversed at 65 °C for about 20 h in the presence of EDTA, proteinase K (NEB, P8107S), and 0.5% SDS. The DNA was then extracted twice with phenol/chloroform/isoamylalcohol (25:24:1) (PCI) (Fisher Scientific, BP1752I-400), precipitated with ethanol, and resuspended in 40 μl of 0.1× TE buffer. Biotin from non-ligated ends was removed using T4 polymerase (NEB, M0203L) (4 h at 20 °C), followed by the extraction with PCI. The DNA was then sheared by sonication for 12 min with 20% amplitude using a Qsonica Q800R2 water bath sonicator. The sheared DNA was used for library preparation with the NEBNext UltraII kit (E7645). Biotinylated DNA fragments were purified using 5 μl streptavidin beads (Fisher Scientific, 65001). DNA-bound beads were used for PCR in a 50 μl reaction for 14 cycles. PCR products were purified using Ampure beads (Beckman, A63881) and sequenced at the Indiana University Center for Genomics and Bioinformatics using NextSeq500 or NextSeq2000. Paired-end sequencing reads were mapped to the genome of *B. subtilis* PY79 (NCBI Reference Sequence NC_022898.1) using the same pipeline described previously[35]. The genome was divided into 10-kb bins. Subsequent analyses and visualization were done using R scripts.

### ChIP-seq

Chromatin immunoprecipitation (ChIP) was performed as described previously[35]. Briefly, cells were crosslinked using 3% formaldehyde (Sigma, F8775-500ML) for 30 min at room temperature and then quenched using 125 mM glycine (Fisher Scientific, 02-002-947), washed using PBS, and lysed using lysozyme. Crosslinked chromatin was sheared to an average size of 170 bp by sonication using Qsonica Q800R2 water bath sonicator. The lysate was precleared using Protein A magnetic beads (GE Healthcare/Cytiva 28951378) and was then incubated with anti-SMC antibodies[70] overnight at 4 °C. The next day, the lysate was incubated with Protein A magnetic beads for one hour at 4 °C. After washes and elution, the immunoprecipitate was incubated at 65 °C overnight to reverse the crosslinks. The DNA was further treated with RNaseA (Promega, A7973), Proteinase K (NEB, P8107S), extracted with PCI (Fisher Scientific, BP1752I-400), resuspended in 100 μl EB (Qiagen), and used for library preparation with the NEBNext UltraII kit (E7645). The library was sequenced at the Indiana University Center for Genomics and Bioinformatics using NextSeq500 or NextSeq2000. The sequencing reads were mapped to the genome of *B. subtilis* PY79 (NCBI Reference Sequence NC_022898.1) using CLC Genomics Workbench (CLC Bio, QIAGEN). Sequencing reads from each ChIP and input sample were normalized by the total number of reads. The ChIP enrichment (ChIP/Input) was plotted and analyzed using R scripts. For comparisons of experimental ChIP enrichment profiles to the ones generated from simulations, the experimental data were filtered and plotted as described in the Supplementary Method section "Filtering of ChIP-seq and MFA data for goodness-of-fit calculations and visual comparisons".

### Whole genome sequencing (WGS)

Cells were grown and collected at the indicated time points under the desired conditions. Collected cells were crosslinked with 3% formaldehyde (Sigma, F8775-500ML) at room temperature for 30 min,

then quenched with 125 mM glycine (Fisher Scientific, 02-002-947). Cells were lysed using Ready-Lyse Lysozyme (Lucigen, R1802M) and treated with 0.5% SDS. Cell lysate was incubated at 65 °C overnight to reverse the crosslinks. The DNA was further treated with RNaseA (Promega, A7973), Proteinase K(NEB, P8107S), extracted with PCI (Fisher Scientific, BP1752I-400), and resuspended in 200 μl of 0.1× TE buffer. The extracted DNA was sonicated using a Qsonica Q800R2 water bath sonicator, prepared using the NEBNext UltraII kit (E7645), and sequenced at the Indiana University Center for Genomics and Bioinformatics using NextSeq500 or NextSeq2000. The reads were mapped to the genome of *B. subtilis* PY79 (NCBI Reference Sequence NC_022898.1) using CLC Genomics Workbench (CLC Bio, QIAGEN). The mapped reads were normalized by the total number of reads. Plotting and analyses were performed using R scripts. For comparisons of experimental WGS profiles to the ones generated from simulations, the experimental data were filtered and plotted as described in the Supplementary Method section "Filtering of ChIP-seq and MFA data for goodness-of-fit calculations and visual comparisons".

**Fluorescence microscopy and quantification of replisome foci**
Cells were grown in the conditions indicated in the experimental timeline shown in Fig. 3a (top). To prepare samples for imaging, 0.2 OD unit of cells was collected, spun down at 6000 rpm for 30 s, then resuspended in 10 μl of PBS solution containing 3 μg/ml of FM4-64 (Molecular Probes) for staining cell membranes and 2 μg/ml of DAPI (Molecular Probes) for staining DNA. 2.5 μl of cells were spotted on a glass slide for imaging. Fluorescence microscopy was performed on a Nikon Ti2E microscope equipped with a Plan Apo 100×/1.4NA phase contrast oil objective and an sCMOS camera. Sample preparations and imaging were performed at room temperature. The number of DnaX-YFP foci was manually quantified using the counting tool in NIS Elements analysis software. 883 to 1457 nucleoids were analyzed at each time point.

**Reporting summary**
Further information on research design is available in the Nature Portfolio Reporting Summary linked to this article.

## Data availability
The Hi-C, ChIP-seq, and WGS data generated in this study have been deposited in NCBI's Gene Expression Omnibus and are accessible through GEO Series accession number GSE282455. Unprocessed microscopy images are available at Mendeley data (https://data.mendeley.com/datasets/7hrspzt89y/1). Any additional information required to analyze the data reported in this paper is available from the corresponding author upon request. Source data are provided with this paper.

## Code availability
The codes used to simulate the SMC-replisome interactions were deposited to GitHub (https://github.com/hbbrandao/bacterialSMCtrajectories)[71].

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

## Acknowledgements

We thank the Wang lab for support and stimulating discussions, Leonid Mirny for computing resources, Xheni Karaboja for technical assistance, Alan Grossman for SMC antibodies, and the Indiana University Center for Genomics and Bioinformatics for high-throughput sequencing. Support for this work comes from National Institutes of Health R01GM141242,

R01GM143182, and R01AI172822 (X.W.). This research is a contribution of the GEMS Biology Integration Institute, funded by the National Science Foundation DBI Biology Integration Institutes Program, Award #2022049 (X.W.).

## Author contributions

Q.L. and X.W. designed the study; Q.L. and X.W. constructed plasmids and strains; Q.L. and Z.R. performed Hi-C experiments and analyses. Q.L. performed whole genome sequencing, ChIP-seq, microscopy experiments, and analyses. H.B.B. performed simulations and data analysis. Q.L. and X.W. wrote the manuscript with input from all authors. X.W. supervised the entire study.

## Competing interests

H.B.B. is a full-time employee of Illumina, Inc., and a board member of Research Theory. The remaining authors declare no competing interests.
