## [Transparent Peer Review file · Nature Communications]

Replisomes restrict SMC translocation in vivo

Corresponding Author: Dr Xindan Wang

Version 0:

Reviewer comments:

Reviewer #1

(Remarks to the Author)

The manuscript entitled 'Replisomes restricts SMC-mediated DNA loop extrusion in vivo' by Qin Liao and collaborators proposes a smart approach based on genetics, genomics, and simulations to study the impact of SMC and replisome encounters on chromosome organization in *B. subtilis*. By manipulating the position of parS sites, controlling the replication process, and using simulations to predict possible scenarios, the authors conclude that collisions between SMC and the replication fork affect the translocation of SMC complexes, likely promoting SMC unloading in most cases, with occasional bypassing. Altogether, the work is solid, supports the conclusions, it is technically sound, and the results presented will contribute significantly to advances in the field. Below are a few suggestions to improve the clarity of the manuscript, along with some questions.

1. The authors performed several simulations to determine parameters that cannot be identified experimentally. This is a valid approach, but they should better distinguish between results derived from experiments and those predicted using simulations. Additionally, they should be more cautious with their interpretations. For instance, the number of cells with replisomes predicted by simulations is 18%, but this value could be higher or lower in vivo. The same applies to all other simulations presented in the article. It is crucial that the authors clearly state this throughout the manuscript, including the last paragraph of the introduction.
2. The authors state that SMC bypassing occurs in some cases, inferred from the low-frequency inter-arm DNA contacts observed beyond the replication stop (Figure 4B). Since this is an important point, performing an SMC ChIP-seq at time 30 vs. time 0 would help support these findings
2. Can the authors provide more details on how the extent of juxtaposition was calculated in Figures 4B to 4D? Was it similarly to Sup Fig 3B? These quantifications should be presented as Sup Fig, since it will facilitate the interpretation of Hi-C maps.
3. Also, they determined the delay time of SMC after collision in different contexts (Figure 4B, lines 297-299; Figure 4D, lines 321-326). Could the authors explain how the delay time was determined? Were these simulations?
4. Could the authors clarify why the threshold for inter-arm contacts in Figures 4C and 4D was set above 0.008? Why was this threshold not applied in Figures 4B or 4E?
5. Based on the Hi-C matrices presented in Figure 4C, the authors estimated that DNA juxtaposition proceeds at a constant speed of ~71 kb/min toward the terminus and ~49 kb/min toward the origin. Could they show how these values were calculated (related to point 2)? Additionally, how can the difference in translocation speed (71 kb/min vs. 49 kb/min) be explained in the absence of replication?

Minor Points :

1. Line 72: In bacteria, there are several types of SMC complexes, and this study focuses only on Smc-ScpAB. It would be more appropriate to introduce Smc-ScpAB explicitly and refer to it as Smc-ScpAB throughout the manuscript, as using "SMC" alone could include other bacterial complexes not loaded by ParB bound to parS.
2. Please indicate how replisome quantification was performed, this information is missing from the method section. Were

cells cross-linked before DAPI staining? If yes, how can there be more cells initiating DNA replication if the cells were cross-linked (see lines 249-251)?

3. When possible, representing the ChIP-seq data for SMC or MFA below each Hi-C plots would improve readability and help in understanding the different Hi-C patterns.

4. In some cases, it is unclear how the limits for SMC enrichment in ChIP-seq experiments were set (e.g., Sup Figure 2). Did the authors use a peak-calling algorithm such as MACS2 to identify SMC-bound regions? This approach might help delineate enrichment zones. Alternatively, representing the ChIP-seq data as $\log_2(\text{ChIP}/\text{input})$, similar to Figure S7D, could make enrichment clearer.

5. There is a problem with the legend of the plot in Sup Figure 3C.

6. Figure 7 is missing descriptions of parS sites, the replicating chromosome, etc.

(Remarks on code availability)

Reviewer #2

(Remarks to the Author)

General comments

In a careful and elegant study, Liao et al describe the progression of replication forks and SMC complexes relative to one another in *B. subtilis*. They find that SMCs have no impact on progression of replication forks. By contrast, they find that such collisions stall SMC complexes, which then either unload, pause or bypass the replisome. Using various experimental tools to control SMC loading and replication fork stalling, the authors compare the consequences of head-to-tail and head-to-head collisions, including collisions that involve one or both replication forks. Using simulations, they determine best parameters for the frequency of bypass versus unloading. Remarkably, the same values are obtained for both orientations, at least within the limits of measurements. These results are solid and will be interesting to the community of SMC researchers and beyond.

The paper is written well, and the simulations are described clearly. Nearly all of my comments are about minor issues of clarity, which if addressed will help readers digest the work. In general, interpretation of these Hi-C maps requires that the reader identify features that are not common in other studies. In line 202 describing figure 2, for example, the reader must identify the portion of arc 1 that has flattened out. A cartoon labeling such a feature and the others, might help. Think of the images that typically accompany a Brewer/Fangman 2-D gel.

Specific comments

1) Line 187-188 – The authors explain that the “gap” could be arise from SMCs that slow or dissociate. It might be more descriptive. I think the authors are saying “The gap is due to complexes delayed by replisome collision relative to those complexes that were already in motion before replication started.”

2) Lines 192-92 – The authors describe how arc 1 extends beyond the replisome. It would help to specify the “leftward replisome”.

3) Line 198 – Would it be more appropriate to say the SMCs collided with the replisome in a head-to-tail manner?

4) Lines 209-211 – I think it would be more precise to say “one SMC complex” since one SMC subunit typically refers to a single polypeptide.

5) Lines 249-251 – Not clear why experiment couldn't have been repeated with formaldehyde fixation to avoid problematic permissive temp imaging.

6) Lines 284-285 and 848 – While the figure legend (line 848) indicates that the replication was synchronized, this was not explicitly stated in lines 284-285. Would help to state explicitly in the text

7) Line 297 – The authors use the 71kb/min measurement from an earlier expt. Can't they determine the rate directly from extension of the second diagonal after replication fork collision, even just a ballpark figure? It is hard to envision why SMCs would travel at a different rate after collision, but it is a formal possibility that the existing data could address.

8) Lines 309 – The asymmetric progression of SMCs (71 vs 49 kb/min) in this experiment indicates that something other than replication biases rates toward/away from origin. It would seem appropriate here to reference Brandao, Wang, Mirny 2019, and whether the data are rigorously consistent with that paper.

9) Line 325 - In figure S3C, SMCs depleted beyond replisome stopping point on leftward side, as described in the text. There is also depletion on roughly a symmetric distance on the rightward side of loading. A reader might conclude that unloading on one side is coupled to unloading on the other, in conflict with figure S7 and lines 349-350. What's up?

10) Lines 334-335 – “Despite some delay” is vague. I think you mean that the collision endpoint lags beyond the non-collision control. Incidentally, some readers might think “endpoint” means completion of a process, when the authors mean it is the last a series of timepoints, even though the process is not done.

11) Line 443 – Is the 24 second value akin to a dwell time. If not it seems like a dwell time behind a stall fork could be derived from these simulations.

12) Lines 463-464 – Figure S7 makes the comparisons between collision with stalled and moving replisomes absolutely clear. There is no way that a naïve reader could reach these conclusions by comparing figs 4E and 6A alone. Citing 4E and 6A first in the text sends the reader on a fruitless mission.

13) Line 587-588 – The authors suggest the replisome evolved to unload SMCs. However, the unloading rate is nearly

identical for the two different replisome orientations. Might these matching parameters mean that there is no special molecular feature of the replisome but instead that unloading is a generic response dictated solely by SMCs upon encounters with obstacles.

(Remarks on code availability)

Reviewer #3

(Remarks to the Author)

The work of Liao et al describes experiments in which the loading of SMC molecules is altered in *B. subtilis* cells, such that replication forks will collide with spreading SMC-complexes, in cells having roughly synchronized round of replications.

The authors show that replication forks are relatively unimpressed if they run into tightly bound/active SMC complexes, and that the latter do get stalled or unloaded from DNA upon encountering of forks.

Specifically, Fig. 1 is a negative result, which is not surprising because replication forks can remove huge objects ahead of them, including large arrays of tightly bound RNA polymerases, so it is not surprising SMC also gets ejected from the chromosome.

Fig. 2 shows that SMC gets affected by bumping into forks. No big surprise.

Figure 3. HPUra stalls replication but does not affect replisome assembly. As was shown before by other groups.

Fig. 4 The authors use an artificial system generating collisions of forks and SMC complexes. Findings are “a majority of SMCs did not bypass the replisome, but were either blocked or removed by the replisome; 2) Although the intensity was faint, the secondary diagonal extended beyond the replisome, suggesting that some SMCs were able to bypass the replisome and continue zipping up the arms;” So what are the new conclusions about SMC functions, if any of the three possibilities (pass, get blocked, get ejected) exist?

Overall, I really don't see how “determining the rules of SMC encounters with replication forks” enhances our understanding of the molecular mechanism of SMC action on chromosomes. Clearly, even in the absence of any *parS* site, the *Bacillus* cells has no problem at all to replicate and segregate its chromosomes, there is no phenotype whatsoever. The work is about learning the obvious, SMC complexes that travel the chromosome actively extruding DNA loops must not hinder replication forks.

Concerning the title: this is completely misleading because nothing at all can be said about SMC activity – DNA loop extrusion – from the experiments performed in the study.

(Remarks on code availability)

its a repository for Hi-C data

Version 1:

Reviewer comments:

Reviewer #1

(Remarks to the Author)

(Remarks on code availability)

Reviewer #2

(Remarks to the Author)

The authors have addressed my concerns, which were basically suggestions on how to communicate their material better. The manuscript is much improved. Excellent in my opinion. The revised version includes important addition that SMCs arriving behind MOVING replication forks stall before the actual collision by an impediment that resolves with time. It will be exciting to see how their future work reveals the culprit (pre-catenanes, Okazaki fragments, etc.) I have only a few minor stylistic suggestions.

1) Lines 190-194 – The authors might mention that SMCs are expressed under their own promoter (or continuous IPTG?) to distinguish this expt from those that precede and follow. Not essential but it speeds the reading.

2) Lines 206-207 - The casual observer might not see curvature in line 1 at $T = 0$, which I think is due to *trnx*. It may help reader to use less descriptive expression, "maintained the shape of line 1 seen in $T = 0$ ", and leave any discussion of *trnx*

where it already sits in lines 343-350.

3) Lines 324-326 – Playing devil's advocate, might the faint arc beyond the HPUra-paused forks be due to SMCs restarting after replication fork collapse in a small fraction of cells. Specifically, might the assays that detect these "bypassers" be more sensitive (measuring the gain of a small signal above zero) than those that measure persistence of replication forks (measuring a small loss from a large signal). If the actual numbers refute this, it might be worth emphasizing.

4) Figure 7A - For consistency, IPTG addition should be annotated in flow chart.

(Remarks on code availability)

Please find our point-by-point responses in blue.

REVIEWER COMMENTS

Reviewer #1 (Remarks to the Author):

The manuscript entitled 'Replisomes restricts SMC-mediated DNA loop extrusion in vivo' by Qin Liao and collaborators proposes a smart approach based on genetics, genomics, and simulations to study the impact of SMC and replisome encounters on chromosome organization in *B. subtilis*. By manipulating the position of parS sites, controlling the replication process, and using simulations to predict possible scenarios, the authors conclude that collisions between SMC and the replication fork affect the translocation of SMC complexes, likely promoting SMC unloading in most cases, with occasional bypassing. Altogether, the work is solid, supports the conclusions, it is technically sound, and the results presented will contribute significantly to advances in the field. Below are a few suggestions to improve the clarity of the manuscript, along with some questions.

We are grateful for the reviewer's encouragement and kind response to our manuscript. We thank the reviewer for their suggestions to improve our manuscript.

1. The authors performed several simulations to determine parameters that cannot be identified experimentally. This is a valid approach, but they should better distinguish between results derived from experiments and those predicted using simulations. Additionally, they should be more cautious with their interpretations. For instance, the number of cells with replisomes predicted by simulations is 18%, but this value could be higher or lower in vivo. The same applies to all other simulations presented in the article. It is crucial that the authors clearly state this throughout the manuscript, including the last paragraph of the introduction.

We have modified the text throughout the manuscript to distinguish experiments from simulations. We have gone through the whole manuscript and worded our interpretations more carefully.

2. The authors state that SMC bypassing occurs in some cases, inferred from the low-frequency inter-arm DNA contacts observed beyond the replication stop (Figure 4B). Since this is an important point, performing an SMC ChIP-seq at time 30 vs. time 0 would help support these findings

We have performed new ChIP-seq experiments to address this point. See new Figure 4c, text Lines 322-324.

2. Can the authors provide more details on how the extent of juxtaposition was calculated in Figures 4B to 4D? Was it similarly to Sup Fig 3B? These quantifications should be presented as Sup Fig, since it will facilitate the interpretation of Hi-C maps.

We have added text (Lines 324-331, 362-367) and figures (new Supplementary Figure 5a-b) to detail these calculations.

3. Also, they determined the delay time of SMC after collision in different contexts (Figure 4B, lines 297-299; Figure 4D, lines 321-326). Could the authors explain how the delay time was determined? Were these simulations?

We have added new figures (new Supplementary Figure 5a-b) for these calculations and addressed this point in the text (Lines 324-331, 362-367).

4. Could the authors clarify why the threshold for inter-arm contacts in Figures 4C and 4D was set above 0.008? Why was this threshold not applied in Figures 4B or 4E?

This labeling was unnecessary. We have removed it for consistency.

5. Based on the Hi-C matrices presented in Figure 4C, the authors estimated that DNA juxtaposition proceeds at a constant speed of ~71 kb/min toward the terminus and ~49 kb/min toward the origin. Could they show how these values were calculated (related to point 2)? Additionally, how can the difference in translocation speed (71 kb/min vs. 49 kb/min) be explained in the absence of replication?

We have added a new figure (new Supplementary Figure 5b) for this calculation. We have discussed this asymmetry in the text (Line 343-350).

Minor Points :

1. Line 72: In bacteria, there are several types of SMC complexes, and this study focuses only on Smc-ScpAB. It would be more appropriate to introduce Smc-ScpAB explicitly and refer to it as Smc-ScpAB throughout the manuscript, as using "SMC" alone could include other bacterial complexes not loaded by ParB bound to parS.

We have added text and citations to clarify this point (Lines 73-79).

2. Please indicate how replisome quantification was performed, this information is missing from the method section. Were cells cross-linked before DAPI staining? If yes, how can there be more cells initiating DNA replication if the cells were cross-linked (see lines 249-251)?

We have added a method section to explain the quantification of replisome foci counting (see maintext Lines 846-856). We have tried multiple ways to crosslink cells before visualizing (see reviewer Figure 1 at the end of this document). However, all the crosslinking methods caused visualization artefacts that affected the quantification of replisome foci. Therefore, we showed the experiments using live cells without crosslinking. We have clarified this point in the text (Lines 266-270).

3. When possible, representing the ChIP-seq data for SMC or MFA below each Hi-C plots would improve readability and help in understanding the different Hi-C patterns.

We have changed our figures accordingly (see new Figures 4 and 5).

4. In some cases, it is unclear how the limits for SMC enrichment in ChIP-seq experiments were set (e.g., Sup Figure 2). Did the authors use a peak-calling algorithm such as MACS2 to identify SMC-bound regions? This approach might help delineate enrichment zones. Alternatively, representing the ChIP-seq data as $\log_2(\text{ChIP}/\text{input})$, similar to Figure S7D, could make enrichment clearer.

We used $\log_2(\text{enrichment ratio})$ plots to determine the limits for SMC enrichment. We have added two figure panels (Figure 4c bottom and Supplementary Figure 2b right) to clarify this point.

5. There is a problem with the legend of the plot in Sup Figure 3C.

Fixed.

6. Figure 7 is missing descriptions of parS sites, the replicating chromosome, etc.

Fixed.

Reviewer #2 (Remarks to the Author):

General comments

In a careful and elegant study, Liao et al describe the progression of replication forks and SMC complexes relative to one another in *B. subtilis*. They find that SMCs have no impact on progression of replication forks. By contrast, they find that such collisions stall SMC complexes, which then either unload, pause or bypass the replisome. Using various experimental tools to control SMC loading and replication fork stalling, the authors compare the consequences of head-to-tail and head-to-head collisions, including collisions that involve one or both replication forks. Using simulations, they determine best parameters for the frequency of bypass versus unloading. Remarkably, the same values are obtained for both orientations, at least within the limits of measurements. These results are solid and will be interesting to the community of SMC researchers and beyond.

We are grateful for the reviewer's encouragement and kind response to our manuscript.

The paper is written well, and the simulations are described clearly. Nearly all of my comments are about minor issues of clarity, which if addressed will help readers digest the work. In general, interpretation of these Hi-C maps requires that the reader identify features that are not common in other studies. In line 202 describing figure 2, for

example, the reader must identify the portion of arc 1 that has flattened out. A cartoon labeling such a feature and the others, might help. Think of the images that typically accompany a Brewer/Fangman 2-D gel.

We thank the reviewer for thoroughly reading our manuscript and suggesting ways to improve clarity. We have modified texts and figures, and addressed all the issues brought up by the reviewer. We think our manuscript is improved because of these changes prompted by the reviewer. We have added a cartoon (new Supplementary Figure 3) to explain the Hi-C features.

Specific comments

1) Line 187-188 – The authors explain that the “gap” could be arise from SMCs that slow or dissociate. It might be more descriptive. I think the authors are saying “The gap is due to complexes delayed by replisome collision relative to those complexes that were already in motion before replication started.”

We have incorporated this suggestion to the text (Lines 201-204).

2) Lines 192-92 – The authors describe how arc 1 extends beyond the replisome. It would help to specify the “leftward replisome”.

Fixed.

3) Line 198 – Would it be more appropriate to say the SMCs collided with the replisome in a head-to-tail manner?

Corrected.

4) Lines 209-211 – I think it would be more precise to say “one SMC complex” since one SMC subunit typically refers to a single polypeptide.

We clarified this point and avoided using “subunit” throughout the manuscript.

5) Lines 249-251 – Not clear why experiment couldn't have been repeated with formaldehyde fixation to avoid problematic permissive temp imaging.

See our responses to Reviewer 1, Minor point 2. We have tried multiple ways to crosslink cells before visualizing (see reviewer Figure 1 at the end of this document). However, all the crosslinking methods caused visualization artefacts that affected the quantification of replisome foci. Therefore, we showed the experiments using live cells without crosslinking. We have clarified this point in the text (Lines 266-270).

6) Lines 284-285 and 848 – While the figure legend (line 848) indicates that the

replication was synchronized, this was not explicitly stated in lines 284-285. Would help to state explicitly in the text

Done.

7) Line 297 – The authors use the 71kb/min measurement from an earlier expt. Can't they determine the rate directly from extension of the second diagonal after replication fork collision, even just a ballpark figure? It is hard to envision why SMCs would travel at a different rate after collision, but it is a formal possibility that the existing data could address.

In this study, we measured that the rate of SMC movement is ~ 71 kb/min at 42°C. We used this number as a “parameter” for experiments in this study (Lines 297-299, 400-407). We did not measure the rate of SMC movement after collision in Figure 4b because the secondary diagonal is very faint, and the estimate of the rate might be inaccurate.

8) Lines 309 – The asymmetric progression of SMCs (71 vs 49 kb/min) in this experiment indicates that something other than replication biases rates toward/away from origin. It would seem appropriate here to reference Brandao, Wang, Mirny 2019, and whether the data are rigorously consistent with that paper.

We have added texts and references to clarify this point (Lines 343-350).

9) Line 325 - In figure S3C, SMCs depleted beyond replisome stopping point on leftward side, as described in the text. There is also depletion on roughly a symmetric distance on the rightward side of loading. A reader might conclude that unloading on one side is coupled to unloading on the other, in conflict with figure S7 and lines 349-350. What's up?

Figure S3C (new Supplementary Figure 5c) indicates that most SMC are unloaded from the collision site. For unloading, both SMC motors are removed from the chromosome, resulting in depletion of SMC enrichment from both sides of the DNA, as was shown for SMC unloading by a specific unloader XerD (Karaboja PMID: 33472056). Old Figure S7 (new Supplementary Figure 9) compares SMC collision with a stalled replisome and a moving replisome. The moving replisome imposes stronger “blocking” to SMC movement. For blocking, only the one SMC motor is affected; the other SMC motor continues translocation (Wang PMID: 28154080; Brandao PMID: 31548377; Brandao PMID: 34312537). In this case, SMC enrichment is only depleted from one side. We have added references and clarified this point in the text (Lines 368-372, 409-413, 520-524).

10) Lines 334-335 – “Despite some delay” is vague. I think you mean that the collision endpoint lags beyond the non-collision control. Incidentally, some readers might think “endpoint” means completion of a process, when the authors mean it is the last a series of timepoints, even though the process is not done.

We have reworded this statement and changed “endpoint” to “end position” throughout the manuscript.

11) Line 443 – Is the 24 second value akin to a dwell time. If not it seems like a dwell time behind a stall fork could be derived from these simulations.

This can be understood as the dwell time of SMC when encountering the replisome. Since we are not discussing dwell time elsewhere, we have decided not to introduce a new term to our system.

12) Lines 463-464 – Figure S7 makes the comparisons between collision with stalled and moving replisomes absolutely clear. There is no way that a naïve reader could reach these conclusions by comparing figs 4E and 6A alone. Citing 4E and 6A first in the text sends the reader on a fruitless mission.

We have changed the citations of relevant figures.

13) Line 587-588 – The authors suggest the replisome evolved to unload SMCs. However, the unloading rate is nearly identical for the two different replisome orientations. Might these matching parameters mean that there is no special molecular feature of the replisome but instead that unloading is a generic response dictated solely by SMCs upon encounters with obstacles.

We have added this point to the text (Lines 688-690) and expanded the discussion on collision orientations (Lines 691-702).

Reviewer #3 (Remarks to the Author):

The work of Liao et al describes experiments in which the loading of Smc molecules is altered in *B. subtilis* cells, such that replication forks will collide with spreading SMC-complexes, in cells having roughly synchronized round of replications.

The authors show that replication forks are relatively unimpressed if they run into tightly bound/active SMC complexes, and that the latter do get stalled or unloaded from DNA upon encountering of forks.

Specifically, Fig. 1 is a negative result, which is not surprising because replication forks can remove huge objects ahead of them, including large arrays of tightly bound RNA polymerases, so it is not surprising Smc also gets ejected from the chromosome.

Fig. 2 shows that Smc gets affected by bumping into forks. No big surprise.

Figure 3. HPUra stalls replication but does not affect replisome assembly. As was shown before by other groups.

Fig. 4 The authors use an artificial system generating collisions of forks and SMC complexes. Findings are “a majority of SMCs did not bypass the replisome, but were either blocked or removed by the replisome; 2) Although the intensity was faint, the secondary diagonal extended beyond the replisome, suggesting that some SMCs were able to bypass the replisome and continue zipping up the arms;” So what are the new conclusions about SMC functions, if any of the three possibilities (pass, get blocked, get ejected) exist?

Overall, I really don't see how “determining the rules of SMC encounters with replication forks” enhances our understanding of the molecular mechanism of SMC action on chromosomes. Clearly, even in the absence of any *parS* site, the *Bacillus* cells has no problem at all to replicate and segregate its chromosomes, there is no phenotype whatsoever. The work is about learning the obvious, SMC complexes that travel the chromosome actively extruding DNA loops must not hinder replication forks.

Concerning the title: this is completely misleading because nothing at all can be said about SMC activity – DNA loop extrusion – from the experiments performed in the study.

We thank the reviewer for their candid evaluation of our manuscript. We respectfully disagree with their assessment that our findings are unsurprising and reveal the obvious. Most of the recent studies investigating SMC activity and function come from in vitro reconstitution experiments. The work presented here seeks to establish whether or not the findings from these in vitro studies reflect the in vivo activity of this important and broadly conserved complex. The *Bacillus subtilis* system is among the most powerful to investigate SMC activity in vivo. While this reviewer felt our experiments were “learning the obvious”, we think others will find our results interesting and surprising. More to the point, we think determining what happens in vivo (surprising or not) represents an important contribution to the field of chromosome biology.

For example, SMC complexes have been shown to translocate along DNA unaffected by many protein-DNA obstacles both in vitro and in vivo. Accordingly, it was unclear whether SMC complexes would be affected by the replisome. Our findings that the replisome restricts SMC translocation in vivo is both novel and unexpected.

The reviewer indicated that our study does not enhance our understanding of the molecular mechanism of SMC action on chromosomes because *B. subtilis* is unaffected by the absence of *parS* sites or ParB. *B. subtilis* cells lacking *parS* sites or ParB can indeed replicate and segregate their chromosomes albeit with an increase in anucleated cells (PMID 16677298). However, this does not preclude using *B. subtilis* to investigate how replisomes manage encounters with SMC complexes and reciprocally how SMC complexes are affected by the replisome.

Finally, related to the title of our manuscript, our previous studies on SMC activity in vivo provide strong evidence that SMC complexes extrude DNA loops (PMIDs: 26253537,

28154080, 34312537). However, to address the concerns raised by the reviewer, we have changed the title to “Replisomes restrict SMC translocation in vivo”.

Reviewer #3 (Remarks on code availability):

its a repository for Hi-C data

As noted in our original “Data and code availability section”, the codes used to simulate the SMC-replisome interactions were deposited to github (<https://github.com/hbbrandao/bacterialSMCtrajectories>); Hi-C, ChIP-seq and WGS data were deposited to the NCBI Gene Expression Omnibus (accession no. GSE282455, reviewer token yponkeygvfmdwd).

Reviewer Figure 1

A. no fixation

B. 3% formaldehyde

C. 4% paraformaldehyde

D. 2.5% glutaraldehyde

Reviewer Figure 1: Fixation methods affected replisome quantification.

Representative images of a *dnaB(ts)* strain (BWX2533, used in Fig. 3a) grown at permissive temperature (30°C, exponential growth) or restrictive temperature (42°C, G1 arrest) and fixed with different fixatives that are commonly used in fixing bacterial cells. The images contain DAPI-stained nucleoid (blue), FM-4-64-stained membrane (red) and YFP-tagged DnaX (green). Scale bar represents 4 µm. Control cells without fixation are shown in **(A)**. By adapting the published fixation protocols (PMID: 25099088; PMID: 21881891; PMID: 33912815; PMID: 39546206), we fixed cells at the sample collection by treating cells with 3% formaldehyde **(B)**, 4% paraformaldehyde **(C)**, or 2.5 % glutaraldehyde **(D)**. A range of fixation conditions including duration of fixative treatment, temperature of treatment, stationary incubation versus rocking incubation, with or without glycine quenching following formaldehyde were systematically tested. All fixation conditions exhibited similar patterns and caused visualization artefacts that affected the quantification of replisome foci. Here we show one set of representative images for each fixative.

Please find our point-by-point responses in blue.

REVIEWER COMMENTS

Reviewer #2 (Remarks to the Author):

The authors have addressed my concerns, which were basically suggestions on how to communicate their material better. The manuscript is much improved. Excellent in my opinion. The revised version includes important addition that SMCs arriving behind MOVING replication forks stall before the actual collision by an impediment that resolves with time. It will be exciting to see how their future work reveals the culprit (pre-catenanes, Okazaki fragments, etc.) I have only a few minor stylistic suggestions.

We thank the reviewer for their praise of our work.

1) Lines 190-194 – The authors might mention that SMCs are expressed under their own promoter (or continuous IPTG?) to distinguish this expt from those that precede and follow. Not essential but it speeds the reading.

We have modified the text.

2) Lines 206-207 - The casual observer might not see curvature in line 1 at $T = 0$, which I think is due to $trnx$. It may help reader to use less descriptive expression, "maintained the shape of line 1 seen in $T = 0$ ", and leave any discussion of $trxn$ where it already sits in lines 343-350.

We have modified the text.

3) Lines 324-326 – Playing devil's advocate, might the faint arc beyond the HPUra-paused forks be due to SMCs restarting after replication fork collapse in a small fraction of cells. Specifically, might the assays that detect these "bypassers" be more sensitive (measuring the gain of a small signal above zero) than those that measure persistence of replication forks (measuring a small loss from a large signal). If the actual numbers refute this, it might be worth emphasizing.

We have modified the text.

4) Figure 7A - For consistency, IPTG addition should be annotated in flow chart.

We have modified the figure.